# Inter- and intra-tumor heterogeneity of metastatic prostate cancer determined by digital spatial gene expression profiling

Lauren Brady [1,5], Michelle Kriner[2,5], Ilsa Coleman[1], Colm Morrissey[3], Martine Roudier[3], Lawrence D. True[3], Roman Gulati [1], Stephen R. Plymate[3,4], Zoey Zhou[2], Brian Birditt[2], Rhonda Meredith[2], Gary Geiss[2], Margaret Hoang[2], Joseph Beechem[2] & Peter S. Nelson [1,3] ✉

Metastatic prostate cancer (mPC) comprises a spectrum of diverse phenotypes. However, the extent of inter- and intra-tumor heterogeneity is not established. Here we use digital spatial profiling (DSP) technology to quantitate transcript and protein abundance in spatially-distinct regions of mPCs. By assessing multiple discrete areas across multiple metastases, we find a high level of intra-patient homogeneity with respect to tumor phenotype. However, there are notable exceptions including tumors comprised of regions with high and low androgen receptor (AR) and neuroendocrine activity. While the vast majority of metastases examined are devoid of significant inflammatory infiltrates and lack PD1, PD-L1 and CTLA4, the B7-H3/CD276 immune checkpoint protein is highly expressed, particularly in mPCs with high AR activity. Our results demonstrate the utility of DSP for accurately classifying tumor phenotype, assessing tumor heterogeneity, and identifying aspects of tumor biology involving the immunological composition of metastases.

[1] Fred Hutchinson Cancer Research Center, Seattle, WA, USA. [2] NanoString Technologies, Inc., Seattle, WA, USA. [3] University of Washington, Seattle, WA, USA. [4] VAPSHCS-GRECC, Seattle, WA, USA. [5] These authors contributed equally: Lauren Brady, Michelle Kriner. ✉email: pnelson@fredhutch.org

Localized prostate cancer (PC) is notable for substantial inter- and intratumor heterogeneity in both phenotype and molecular composition. At the time of diagnosis, biopsies often demonstrate the presence of multiple histological Gleason patterns, and independent cancer foci harbor distinct structural genomic alterations such as those involving TMPRSS2-ERG rearrangements[1–5]. As observed in most solid tumors, PCs are comprised of heterogeneous populations of neoplastic cells interacting within complex ecosystems of resident cell types such as fibroblasts and vascular endothelium, infiltrating cell types including immune cells, as well as nutrients, growth factors, collagens, and other constituents that collectively contribute to an organizational framework that supports cancer cell survival and growth[6]. Notably, the variation in the histological patterns are highly prognostic of PC outcomes[1], and molecular assays of gene expression also associate with recurrences following prostatectomy and radiotherapy, indicating a high degree of interindividual variation in tumor behavior[7].

The recognition that cancers originating from the same organ can harbor a spectrum of oncogenic and tumor suppressor alterations between individuals led to concerted efforts to understand the diversity and frequency of pathogenic molecular features present in common human malignancies. Applying genome-scale technologies to cancer resulted in the construction of The Cancer Genome Atlas (TCGA) that provided a working taxonomy for numerous solid tumors and hematological malignancies. The results of TCGA for localized PC[8] and further efforts involving analyses of metastatic PC identified molecular subtypes of PC with otherwise indistinguishable histological features and nominated new therapeutic targets[9–12]. Collectively, these studies emphasize that PC is driven by a diverse spectrum of oncogenic aberrations and provide strong rationale for personalized/precision approaches for cancer therapy.

While the application of advanced technologies such as laser-assisted microdissection, proteomics, single-cell sequencing, and spatial transcriptomics have delineated the molecular diversity of localized PCs, an understanding of the intraindividual and intratumoral diversity of metastatic PC is lacking[13–17]. Metastatic biopsy-based studies have generally been limited to evaluating a single metastatic site using methods that integrate events amalgamated from a large population of individual malignant and benign cells[9,11]. Autopsy studies capable of evaluating multiple disseminated tumors within an individual patient determined that in the majority of patients, all PC metastasis share a common monoclonal origin, but subsequent therapeutic pressures promote a degree of diversity with respect to resistance mechanisms[10,18,19]. In addition to genomic aberrations, recent studies have identified processes contributing to therapy resistance involving alterations in tumor phenotypes through transdifferentiation that may be driven by epigenetic modifications[20–23]. However, the intra- and intertumor variation in these phenotypes and potential associations with tumor-microenvironment (TME) features such as immune responses has not been evaluated.

In this study, we seek to investigate the inter- and intratumor variation in gene expression using an approach termed digital spatial profiling (DSP) for quantitative, high-plex analysis of mRNAs and proteins in spatially defined regions of PC metastasis[24,25]. We apply this method to the study of formalin-fixed tumor biospecimens from multiple metastatic sites, including bone, acquired through rapid autopsy. In addition to assessing the variation in individual genes encoding molecular targets for specific therapeutics, we also use DSP to categorize tumor phenotypes based on gene expression programs that indicate the activity of androgen receptor (AR) activity, neuroendocrine (NE) differentiation, and FGFR/MEK signaling, as well as the composition of immune cells and immunomodulatory cytokines and chemokines.

## Results

**Digital spatial gene expression profiling of PC metastases.** To characterize the phenotypic heterogeneity and spatial distribution of tumor cells in metastatic PCs (mPCs), we constructed tissue microarrays (TMAs) representing diverse anatomic sites of tumor dissemination in 27 patients with therapy-refractory mPC. The study design included two anatomically distinct metastatic sites per patient to evaluate intraindividual heterogeneity. Three spatially distinct regions were punched from each tumor to evaluate intratumoral heterogeneity. For one patient, four tumors were included to further evaluate the extent of tumor heterogeneity. Tumor samples were collected over an 8-year time interval, formalin-fixed and paraffin-embedded (FFPE) at the time of resection, and stored as tumor blocks. In total, 168 tumor cores from 56 mPC tumors were arrayed and used for subsequent analyses (Fig. 1a).

To quantitate gene expression in spatially defined tumor regions, we assembled a gene panel of utility in assessing the molecular composition of neoplastic disease that included transcripts for the functional classification of PC phenotypes and the categorization of specific cell types. These included genes comprising signatures of AR activity, NE differentiation, proliferation, fibroblast growth factor (FGF), and mitogen activated protein kinase (MAPK) activity, loss of the retinoblastoma gene (RB1), and markers of cell types including macrophages (e.g., CD163), T cells (e.g., CD3E), and B cells (e.g., MS4A1). In total, 2093 unique genes comprised the DSP panel (Supplementary Data File 1). We designed a series of barcoded oligonucleotide probes (median of 10 probes/target mRNA) for each gene of interest for a total of 18,120 probes, including oligos targeting particular gene isoforms such as AR splice variants. For measuring protein expression, our panel included AR, synaptophysin (SYP), and the 55 proteins in the NanoString Human Immuno-Oncology, Drug Target, Activation Status, Cell Typing, and Pan-Tumor panels.

Sections of the TMA were used for histological analysis following hematoxylin and eosin staining. Serial sections were stained simultaneously with fluorescently labeled antibodies specific for the leukocyte markers CD3 and CD45, epithelial cell marker PanCK, and the nuclear stain SYTO 13 for DSP. For each tumor core, one 500 μM region of interest (ROI) was selected, attempting to acquire the largest percentage of tumor cells from each tumor core section. The cellularity of a typical core averaged 1200 cells, though a small number of cores was composed primarily of fat cells or acellular stroma. Each ROI was assessed for the composition of neoplastic cells and annotated based on the cellular composition as pure tumor (T), predominantly tumor (>50% of the sampled area) with the remainder benign cells or stroma (TS), predominantly benign cells or stroma with some tumor (ST), or purely benign cells or stroma (S) (Supplementary Data File 2).

DSP was performed as previously described[25]. Following probe hybridization, UV cleavage, and barcode collection, gene expression was quantitated by Illumina sequencing (for protein) or by PCR amplification and Illumina sequencing (for RNA) (Fig. 1b). Manual inspection of the TMAs used for DSP determined that of 168 cores arrayed, 7 were either missing or were 100% fat and 1 was entirely stroma devoid of tumor cells. For the RNA DSP assay, seven additional ROIs were missing and four did not pass sequencing quality control. These cores were excluded from further analyses for both protein and RNA experiments.

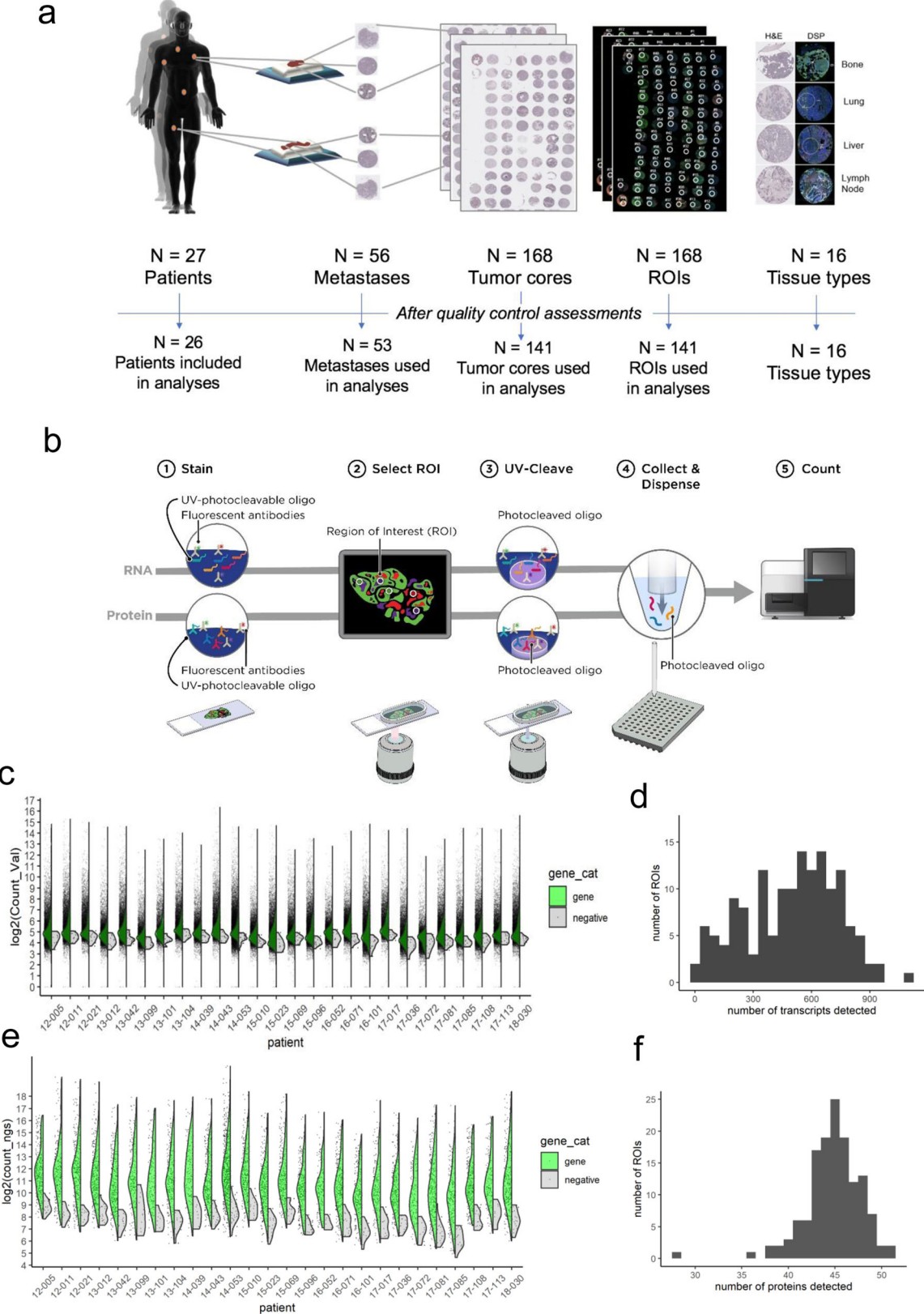

For the RNA DSP, probe counts were collapsed to gene counts by first removing outlier probes and then taking the geometric mean of remaining probes for each gene in each ROI. Gene and negative count distributions for each patient ($N = 6$ ROIs for most patients) are shown in Fig. 1c. Negative probe counts were used to set a limit of quantitation (LOQ), which was defined as the geometric mean plus two standard deviations of the negative probes. The median number of genes above LOQ was 552 per ROI (range 18–1088) (Fig. 1d) (Supplementary Data File 3). The distributions of protein

**Fig. 1 Digital spatial profiling of archived formalin-fixed paraffin-embedded prostate cancer metastases. a** Schematic of study participants. $N = 27$ patients from the UW Rapid Autopsy Program were selected with two sites of metastasis per patient ($N = 56$). One 500 μM region of interest (ROI) per core was selected for DSP. Of the total 168 ROIs, 141 were utilized for analysis. Reasons for exclusion include missing/100% fat ($N = 14$), 100% stroma ($N = 1$), quality control failure ($N = 4$), or $\leq$100 genes detected ($N = 8$). **b** Schematic of multianalyte DSP workflow. Serial sections of the TMA were run through GeoMx RNA (top) or GeoMx protein (bottom) assays. Both assays were read out by next-generation sequencing. **c** Split violin plot overlaid on scatterplot of RNA assay counts by patient ($N = 3$-$12$ ROIs for most patients). Gene counts are displayed on the left side (green) and the geometric mean of negative probe counts is displayed on the right side (gray). **d** Histogram of the number of genes above the limit of quantitation (LOQ) per ROI for the RNA assay. LOQ was defined as the geometric mean of the negative probes × geometric standard deviation of negative probes squared. **e** Split violin plot overlaid on scatterplot of protein assay counts by patient ($N = 3$-$12$ ROIs for most patients). Antibody counts are displayed on the left side (green) and the geometric mean of negative control antibodies (mouse and rabbits IgGs) displayed on the right side (gray). **f** Histogram of the number of proteins above the limit of quantitation (LOQ) per ROI. LOQ was defined as three times the geometric mean of the negative control antibodies.

counts for each patient are shown in Fig. 1e. LOQ was defined as three times the mean of counts of the negative control antibodies (mouse and rabbit IgGs). The median number of proteins above LOQ per ROI was 17 (range 8–35) (Fig. 1f). We removed eight ROIs with $\leq$100 genes detected in the RNA DSP assay from both protein and RNA datasets, leaving 141 ROIs from 53 metastases (26 patients) for further analyses (Supplementary Data File 4).

**Quantitative gene expression measurements from spatially defined regions of PC metastases identify distinct phenotypes.** mPC exhibits a number of notable features including the expression of a cell differentiation, survival, and proliferation program regulated by the AR[26–28]. Therapeutic pressures designed to repress AR signaling can promote resistance pathways involving transdifferentiation to small cell NE phenotypes (SCNPC)[22]. These tumors express a spectrum of NE genes such as SYP and chromogranin (CHGA). We used DSP-based quantitation to assess the activity of these pathways across 53 metastasis. For each of the tumors used for DSP profiling, we previously generated whole transcriptome RNAseq measurements of gene expression using frozen tumor tissue. To facilitate comparisons with tumor expression profiles obtained by RNAseq, we initially averaged the DSP measurements from all three cores/ROIs from the same tumor.

Overall, we observed substantial intertumor heterogeneity across patients with wide ranges of expression and pathway activity between tumors (Fig. 2a). Notably, the DSP-based assessments of AR- and NE-activity scores matched the bulk RNAseq-based measurements with high concordance, $r = 0.83$ and $r = 0.69$, respectively (Fig. 2b, c), as were the DSP and RNAseq measures of cell cycle progression (CCP) scores ($r = 0.67$; $p < 0.0001$) (Fig. 2d) while a signature of FGFR-MEK signaling ($r = 0.16$) (Fig. 2e) was not significantly correlated.

We have previously determined that mPC can be broadly partitioned into six phenotypic categories based on the activity of AR and NE programs: AR+/NE−; AR^low^/NE−; AR−/NE−; AR−/NE^low^; AR+/NE+; and AR−/NE+[23,29]. Each phenotype is classified as follows: AR+/NE− tumors are defined by positive expression of AR-regulated genes, also referred to as AR signature genes, and lack of expression of NE associated genes; AR^low^/NE− tumors are comprised of weak or heterogeneous expression of AR-regulated genes and a lack of expression of NE associated genes; AR−/NE− tumors are determined by lack of expression of both AR signature genes and NE associated genes; AR−/NE^low^ tumors are defined by lack of expression of AR signature genes and low or heterogenous expression of NE associated genes; and AR+/NE+ tumors coexpress genes indicating AR and NE pathway activity. We next used the DSP expression measurements of 23 AR and NE genes to perform multidimensional scaling (MDS) to assign phenotypes to the profiled tumors (Fig. 2f). The classification of phenotypes matched the bulk

RNAseq-based assignment with 46 concordant and 7 discordant classifications (Fig. 2a).

In addition to the cellular pathways and expression programs that define phenotypes, several individual genes and their encoded proteins play important roles in PC pathobiology and serve as targets for therapeutics. These targets include *FOLH1* which encodes prostate-specific membrane antigen (PSMA), where PSMA-conjugated radioligands are being evaluated for imaging as well to focally direct therapeutic doses of radioisotopes[30–32]. The histone methyltransferase enhancer of zeste homolog 2 (*EZH2*) is involved in cellular reprogramming/transdifferentiation and may contribute to AR-directed therapy resistance[33,34]. The antiapoptotic protein BCL2 and checkpoint kinase inhibitor Wee1 are upregulated in SCNPCs and therapeutics directed toward these proteins inhibit SCNPC growth in preclinical models[35]. DSP quantitation of each of these targets demonstrated that interindividual heterogeneity with measurements closely aligned with RNAseq-based levels (Fig. 2g–j). Notably, the expression variation between patients suggests that a precision approach may be required to establish efficacy specifically in patients most likely to benefit by virtue of target expression.

**DSP identifies limited intraindividual diversity in metastatic PC phenotypes.** Most patients with mPC have multiple sites of tumor dissemination that may include lymph nodes, bone, and various soft tissues such as liver, adrenal, and lung. The similarities and differences in the phenotypes and genotypes between the disseminated tumor sites will impact the utility of sampling any individual site for overall tumor classification and the performance of a predictive biomarker for a given therapy. We next compared the DSP-based phenotypic classification of metastasis within each patient and also evaluated expression programs indicating cell proliferation (CCP) status, FGF/MAPK activity, and *RB1* loss. Overall, there was high concordance in the phenotype call within a given patient with 82% of randomly sampled pairs of tumor ROIs from the same patient having the same phenotype classification (Fig. 3a). In contrast, only 54% of ROIs were phenotypically concordant when randomly comparing ROIs across all tumors and all patients. The scores for pathway activities were also generally concordant between metastasis from the same patient, with substantially greater diversity across patients (Fig. 3 and Fig. S1). However, there were notable exceptions: in seven patients, tumors were classified into different phenotype categories. For example, patient 15-096 had one tumor classified as AR+/NE+, whereas a separate metastasis was AR+/NE− (Fig. 3b–d). Patient 12-011 had one tumor classified as AR+/NE− and one tumor AR+/NE+ and patient 15-010 had one tumor classified as AR^low^/NE− and one tumor AR−/NE+ (Fig. 3e, f). At the individual gene level, there was clear evidence of divergent gene expression for transcripts that comprise NE differentiation, for example, SYP, ASCL1, and ONECUT2. The

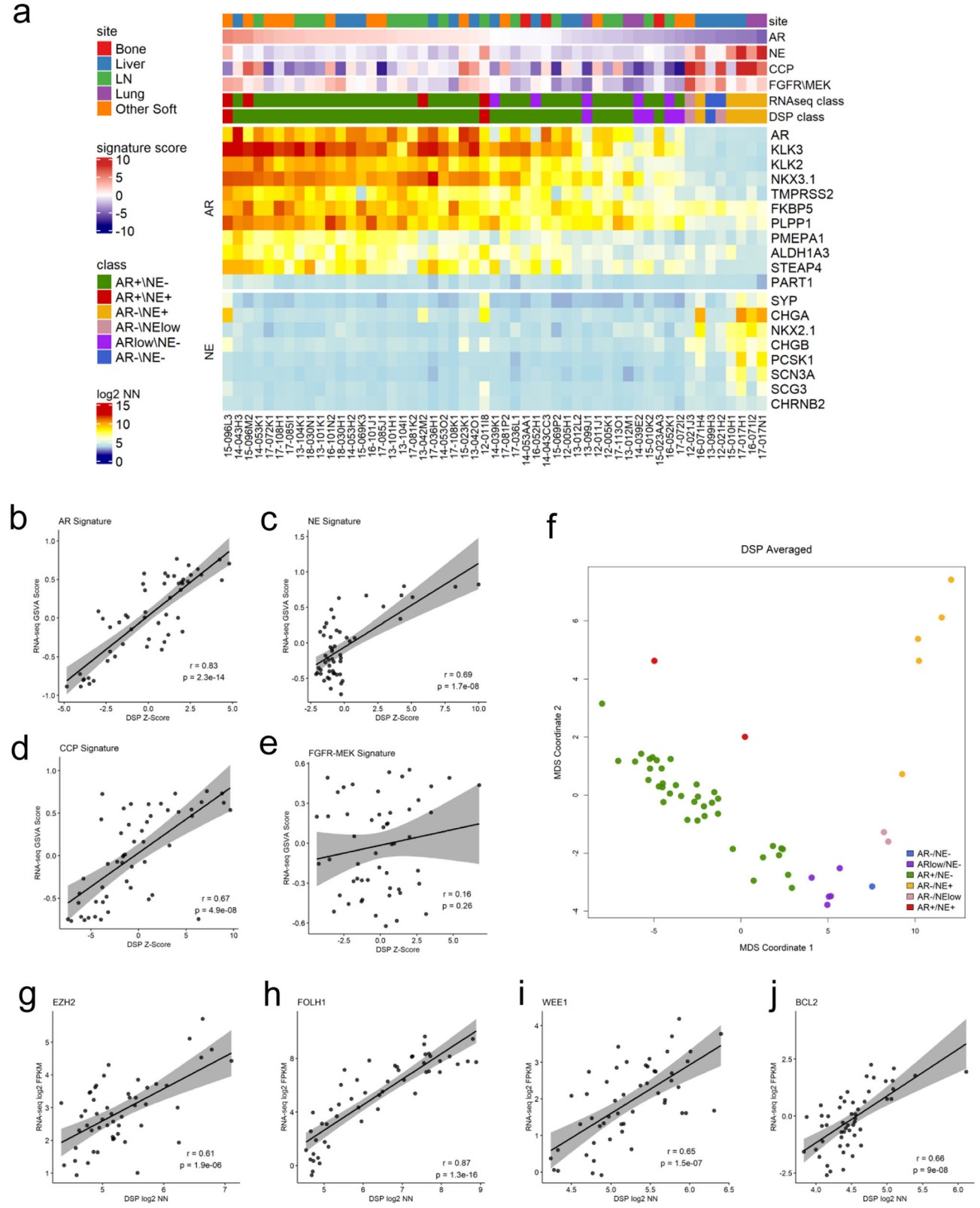

intertumor heterogeneity included transcripts encoding proteins of potential utility in therapeutics, such as the cell surface protein NCAM1, to which antibodies and CAR-T cells have been developed (Fig. 3e, f)[36]. Though unusual, these findings indicate that therapy resistance can occur via distinct mechanisms and may require combinations of treatments to address diverse drivers of progression within an individual patient.

PC is notable for a high predilection to disseminate to bone and produce a spectrum of osteolytic and osteoblastic bone responses. Using RNA-based methods to assess gene expression in bone has been challenging due to formic acid decalcification procedures that enable tissue sectioning but generally degrade nucleic acids. To evaluate the utility of DSP in assessing transcript levels in PC bone, we sampled one bone metastasis and one

**Fig. 2 DSP classifies mPC subtypes and quantitates the expression of therapeutic targets. a** Heatmap of DSP gene expression correlated with bulk RNAseq across androgen receptor (AR), neuroendocrine (NE), cell cycle progression (CCP), and FGFR/MEK gene signatures ($N = 141$ ROIs averaged to 53 tumors from 26 patients). Results are expressed as mean gene signature $Z$-scores and mean log$_2$ negative-normalized (NN) gene expression and presented according to color scales. RNAseq class and DSP class are the phenotypes assigned to the samples using each dataset. **b-e** Scatterplots comparing gene expression of RNAseq GSVA scores to mean DSP $Z$-scores across AR, NE, CCP, and FGRF-MEPK gene signatures ($N = 141$ ROIs averaged to 53 tumors from 26 patients.) Two-sided test for association using Pearson's correlation coefficient, $r$; $p$ value shown on plots. **f** Multidimensional scaling (MDS) plot of mCRPC phenotypes as defined by DSP using the mean DSP log$_2$ negative-normalized expression of 23 AR and NE genes ($N = 141$ ROIs averaged to 53 tumors from 26 patients.). **g-j** Scatterplot comparison of single genes EZH2, FOLH1, WEE1, and BCL2 mean DSP log$_2$ negative-normalized expression ($N = 141$ ROIs averaged to 53 tumors from 26 patients) vs RNAseq log$_2$ FPKM ($N = 53$ tumors from 26 patients.) Two-sided test for association using Pearson's correlation coefficient, $r$; $p$ value shown on plots.

paired soft tissue metastasis from each of three patients (Fig. S2a) and compared the gene expression output across the 2104 genes assayed. The bone samples underwent standard decalcification procedures prior to formalin fixation, paraffin embedding, and tissue coring for TMA construction. For the nine ROIs comprising the three soft tissue metastasis, the mean count was $2.4 \times 10^5$ (range $7.5 \times 10^4$ to $5.0 \times 10^5$) compared to a mean count of $9.3 \times 10^4$ (range $4.2 \times 10^4$ to $1.4 \times 10^5$) for the nine ROIs from bone metastasis. The number of genes detected above the LOD was on average 731 (range 400–1088) in soft tissue cores vs 456 (range 216–675) in bone (Fig. S2b, c). Despite overall lower probe counts per gene, of the genes detected, there was a high concordance in transcript levels for the pairs of soft tissue and bone metastasis ($R^2 = 0.90$ for 14-043; 0.78 for 14-053; and 0.43 for 15-023) (Fig. S2d), and each classified as AR+/NE− phenotype (Fig. S2e). In addition to PC-specific transcripts, a comparison of bone ROIs vs paired soft tissue ROIs identified microenvironment-specific gene expression such as *SPP1* and *IBSP* from bone (Fig. S2f).

**DSP identifies a subset of metastases with intratumoral phenotypic heterogeneity.** An attribute of the DSP technology is the ability to quantitate transcript and protein levels in defined regions of a tumor that include selected foci within tumor-rich areas, peritumoral margins, and stroma. Our study design focused on intratumoral heterogeneity defined first by a spatial dimension comprising TMA cores and a second intratumoral spatial dimension determined by a 500-μM-sized ROI for UV-directed probe cleavage and capture.

To assess intratumoral heterogeneity, we compared the transcript measurements between three spatially distinct ROIs obtained from each metastasis. A histological review of the ROIs selected for DSP analysis classified each ROI into one of four categories based on tumor cell composition and cellularity as pure tumor (T), predominantly tumor (>50% of the sampled area) with the remainder benign cells or stroma (TS), predominantly benign cells or stroma with some (<50%) neoplastic cells (ST), or purely benign cells or stroma (S). As our study design focused on sampling intratumoral ROIs, only one ROI classified as (S), and this was not included in further analyses of tumor gene expression heterogeneity. Overall, there was high intratumor concordance for the transcript-defined tumor phenotypes (Fig. 4a) with 96% of randomly sampled pairs from the same tumor having the same phenotype classification. At the level of individual genes, the concordance of transcript abundance from ROIs from the same tumor exceeded transcript concordance derived from a separate metastatic tumor from the same patient, with further divergence observed when comparing measurements from tumors across individuals (Fig. 4b).

Although intratumoral ROIs were generally concordant in classifying phenotypes and the status of particular signaling programs, there were notable outliers that identified heterogeneity within a given tumor with implications for mechanisms of

therapy resistance. For two tumors, the tumor phenotype classification diverged across ROIs, with one tumor having ROIs with both AR+/NE− and AR$^{low}$/NE− regions (12-005K1) and one tumor having ROIs with both AR$^{low}$/NE− and AR−/NE− phenotypes (15-010K2) (Fig. 4a). Patient 12-005K1 is highlighted as an example of this intratumor heterogeneity (Fig. 4c) comparing the AR$^{low}$/NE− ROI to the AR+/NE− ROIs. Reduced expression of AR associated genes (e.g., *KLK3*) in this analysis is consistent with an AR$^{low}$ phenotype. Though generally highly concordant, in several tumors the assessments of AR, NE, and cell cycle activity from different ROIs within the same tumor also diverged (Fig. 4d).

We have previously reported that a subset of metastatic PCs resisting AR-directed therapy expresses both AR activity and NE activity and these cancers are classified as amphicrine tumors[23]. Using methods that sample bulk RNA such as RNAseq cannot distinguish whether the tumor cell population comprises a homogenous population of neoplastic cells that individually express both programs, true amphicrine cells, or whether the tumor mass is comprised of heterogenous cells with foci of AR +/NE− and foci of AR−/NE+ cells and potentially other cell types. For patient 15-096, bulk RNAseq classified both 15-096L and 15-096M metastasis as AR+/NE+, whereas by DSP, each ROI from 15-096M2 classified as AR+/NE− but all ROIs from 15-096L3 classified as AR+/NE+. Further, when evaluating scores for CCP, FGFR/MAPK, and RB1 loss activity, as well as individual genes comprising the AR and NE scores, modest variation was observed across the individual ROIs (Fig. 4e).

To assess intratumoral heterogeneity more deeply, we took a full-face section of the 15-096M1 lymph node metastasis, selected 12 circular ROIs of 200–500 μM in diameter, and quantitated transcript levels by DSP. While the full section was comprised of densely-populated neoplastic cells throughout (Fig. 5a), one region, comprising ~10% of tumor area, was PanCK-positive, whereas the remainder was negative for PanCK immunoreactivity (Fig. 5b). Multiple ROIs from the PanCK-positive region were classified as AR+/NE− by gene expression (Fig. 5c, d). ROIs from regions spatially distant from the PanCK-positive cells were classified as AR−/NE+ or AR−/NE−, with lower expression of AR and AR-regulated genes such as TMPRSS2, and increased expression of genes associated with transdifferentiation, cell plasticity, and proliferation such as EZH2 and Ki67 (Fig. 5c–k).

**AR-V7 expression varies within and across PC metastases.** Alternative splicing of gene transcripts occurs commonly in carcinomas. Notably, in PC, several splice variants of the AR have been identified, particularly in the context of resistance to ADT, and specific splice variants, such as AR-V7, may promote resistance to second generation AR pathway inhibitors (ARSi)[37,38]. We determined the presence of AR-V7 in each tumor by transcript reads spanning canonical and cryptic exons determined by bulk RNAseq and found that for most patients (25/26; 96%), both tumors evaluated were either concordantly AR-V7+ or AR-V7−

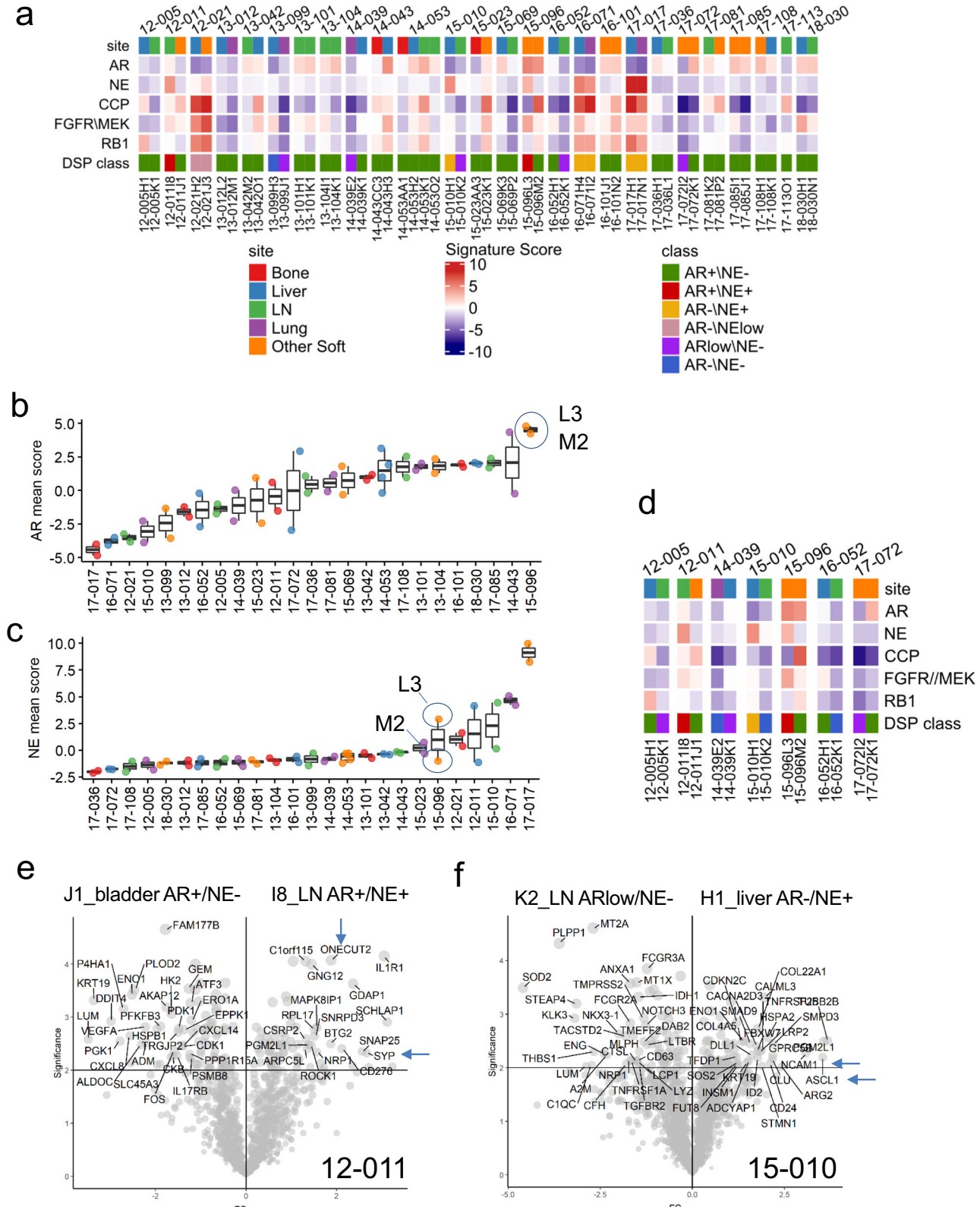

(Fig. 6a). For two patients, 16–052 and 17–081, tumors were discordant with respect to AR-V7 status. We next used DSP to further evaluate the intratumoral heterogeneity of AR-V7. We designed a series of barcoded oligonucleotide probes with specificity for each of the exons comprising the full-length *AR* gene and cryptic exon 3 (CE3) that comprises the AR-V7 transcript (Fig. 6b). Notably, when evaluating AR-V7 expression in the

individual ROIs by DSP, we observed more substantial heterogeneity: regions from the same tumor expressed high AR-V7, while other regions lacked detectable AR-V7 transcript (Fig. 6a). Further, for six tumors there was discordance between AR-V7 status measured by DSP and bulk RNAseq (Fig. 6a–d). To further assess AR-V7 expression, we evaluated AR-V7 protein by immunohistochemistry of the TMA cores. Overall, there was a

**Fig. 3 DSP identifies intertumoral heterogeneity in prostate cancer phenotypes. a** Heatmap of 141 ROIs averaged to 53 tumor cores grouped by 26 patients highlighting androgen receptor (AR), neuroendocrine (NE), cell cycle progression (CCP), FGFR/MEK, and RB1 gene signatures. Mean gene signature Z-scores are shown according to color scale. Data graphed as boxplots indicating differences in **b** AR gene signature and **c** NE gene signature mean Z-score across 138 ROIs averaged from 52 sites from 25 patients with at least two tumors included on the TMAs. Blue circles highlight similar AR signature expression and differential NE signature expression in two different sites of metastasis in patient 15-096. L3 periaortic, M2 diaphragm. Boxes represent the median and interquartile range (IQR) and the upper and lower whiskers extending to the values that are within 1.5 × IQR; data beyond the end of the whiskers are outliers and plotted as points. **d** Heatmap of 35 ROIs averaged to 14 tumor cores from seven discordant patient samples adapted from **a**. Results are expressed as mean gene signature Z-scores and presented according to color scale in **a**. **e** Volcano plot demonstrating intrapatient heterogeneity in individual 12-011. Two sample sites, bladder and lymph node (LN) with different mCRPC phenotypes were compared ($N = 3$ regions of interest (ROIs) per site) and genes associated with a NE phenotype are enriched in the sample I8_LN AR+/NE+ when compared to J1_bladder AR+/NE−. **f** Volcano plot demonstrating intrapatient heterogeneity in individual 15-010. Two sample sites, lymph node and liver with different mCRPC phenotypes were compared ($N = 3$ ROIs per site). Genes associated with AR signature were enriched in sample K2_LN AR$^{low}$/NE− and genes associated with a NE phenotype are enriched in the sample H1_liver AR−/NE+.

significant positive correlation between AR-V7 IHC and transcript levels measured by DSP ($r = 0.54$, $p < 0.0001$, Fig. S3g) and most tumors demonstrated homogenous absence or presence of AR-V7 nuclear staining. However, there were several tumors where AR-V7 expression was heterogenous by IHC (Fig. S3f), a finding concordant with the high and low AR-V7 quantitation by DSP that varied by ROI within a tumor. We note that a degree of divergence can be expected across these measurements as the bulk RNAseq was obtained from a different portion of the metastatic tumor compared to the selection of FFPE embedded tumor cored for TMA construction. Further, the AR-V7 IHC was performed on a section of the TMA approximately ten sections, equating to ~50 μM, from the section used for DSP analyses.

**Immune cell types and checkpoint protein expression varies across PC metastases.** Metastatic PC is notable for the general lack of response to immune-based therapeutics including those designed to block immune checkpoints such as anti-CTLA4, PD1, and PD-L1 antibodies. The immune cell repertoire of localized PC has been characterized, but other than PD-L1 expression, which is generally quite low, the inflammatory cell composition and the expression of various cytokines, chemokines and other immune modulatory proteins has not been well-characterized in metastatic PC[39–41]. To evaluate the presence of immune cell populations, we used DSP to quantitate transcripts encoding proteins that mark distinct immune cell types including T cells, B cells, macrophages, neutrophils, dendritic cells, NK cells, myeloid-derived suppressor cells, as well as a spectrum of chemokines, cytokines, and checkpoint proteins. The ROIs captured in this study focused on regions enriched for neoplastic cells, and overall, these tumor cell rich regions were largely devoid of immune cells of any phenotype (Fig. 7a). We confirmed this observation by manual counts of CD3-positive leukocytes in each tissue section and ROI. The mean number of CD3+ cells was 13.9 per ROI (range 0–250) (Fig. 7b; Supplementary Data File 2). Overall, macrophages were the most commonly detected cell type with 69 of 141 tumor ROIs positive for CD68 by DSP and 48 of 141 for CD163 (Fig. 7a). Markers of CD4 or CD8 T cells were very infrequently detected.

To confirm the findings derived from transcript-based measurements, we also used DSP to quantitate protein abundance using a multiplexed panel of 57 antibodies with three control anti-mouse and anti-rabbit antibodies (Fig. 7c). We confirmed high concordance between DSP–RNA and DSP protein for several well-established genes known to exhibit differential expression based on PC phenotypes: *AR*, *Ki67*, and *SYP* (Fig. S3). The DSP protein assessments confirmed the overall lack of intratumoral cells expressing CD3, CD4, or CD8 with relatively higher levels of CD68 (Fig. 7c).

The paucity of intratumoral immune cells prompted a further analysis of possible mechanisms contributing to a deficient immune response. Previous studies determined that the immune checkpoint protein PD-L1 is rarely expressed in prostate carcinoma, either localized or metastatic[39]. We confirmed this result as PD-L1 protein levels were not detectable above background in any tumor or tumor ROI (Fig. 7c, d). The expression of other checkpoint proteins for which therapeutic antibodies are approved, CTLA4 and PD1, were similarly below measurable levels in >90% of ROIs, either by transcript or antibody-based measurements (Fig. 7c, d).

Recently, additional molecules that influence immune cell activation have been identified including LAG3, TIM-3, TIGIT, VISTA, B7-H3, BTLA, and others[42,43]. We evaluated the expression of these immune checkpoint targets using DSP-based quantitation of transcript and/or protein levels and found high expression of B7-H3/CD276 in 25 of 53 tumors at the transcript level, and in 50 (88%) of the tumors by protein analysis. Though B7-H3 levels were readily detectable in tumors of all CRPC phenotypes, the highest expression was consistently observed in the AR+/NE− subtype. Further, TIM-3 was also expressed highly in more than 37% of tumors, and expression correlated strongly with B7-H3 (Fig. 7c and Fig. S3).

## Discussion

PCs, as with most other human malignancies, have generally been categorized by histomorphology, but can now be subtyped based on gene expression profiles, genomic aberrations, and/or molecular features of tumor microenvironments[9,21,23,29,41,44]. Critically, molecular classification may point toward therapeutic strategies that are likely to result in improved outcomes, or conversely avoid treatments where resistance is likely to preexist or emerge rapidly[45]. Carcinoma of the prostate is a representative example of the molecular complexity that underlies tumor behavior with numerous recurrent genomic and epigenomic aberrations that drive tumor development, progression to metastasis, and the emergence of treatment resistance[9,11,12]. To accurately classify subtypes of PC, multiplexing methods are required that can determine the molecular states of numerous genes or gene products simultaneously, ideally using standard pathology workflows. However, tumor heterogeneity may influence accurate tumor classification[46–48]. Further, insights with respect to the importance of tumor–host interactions are emerging through detailed studies of tumor cells with the immune system and other microenvironment components[49]. New technologies such as spatial transcriptomics, multiplexed immunofluorescence, CODEX, and mass cytometry are capable of integrating multiplexed molecular assays with a spatial context which can detect heterogeneity and may enhance the accuracy of

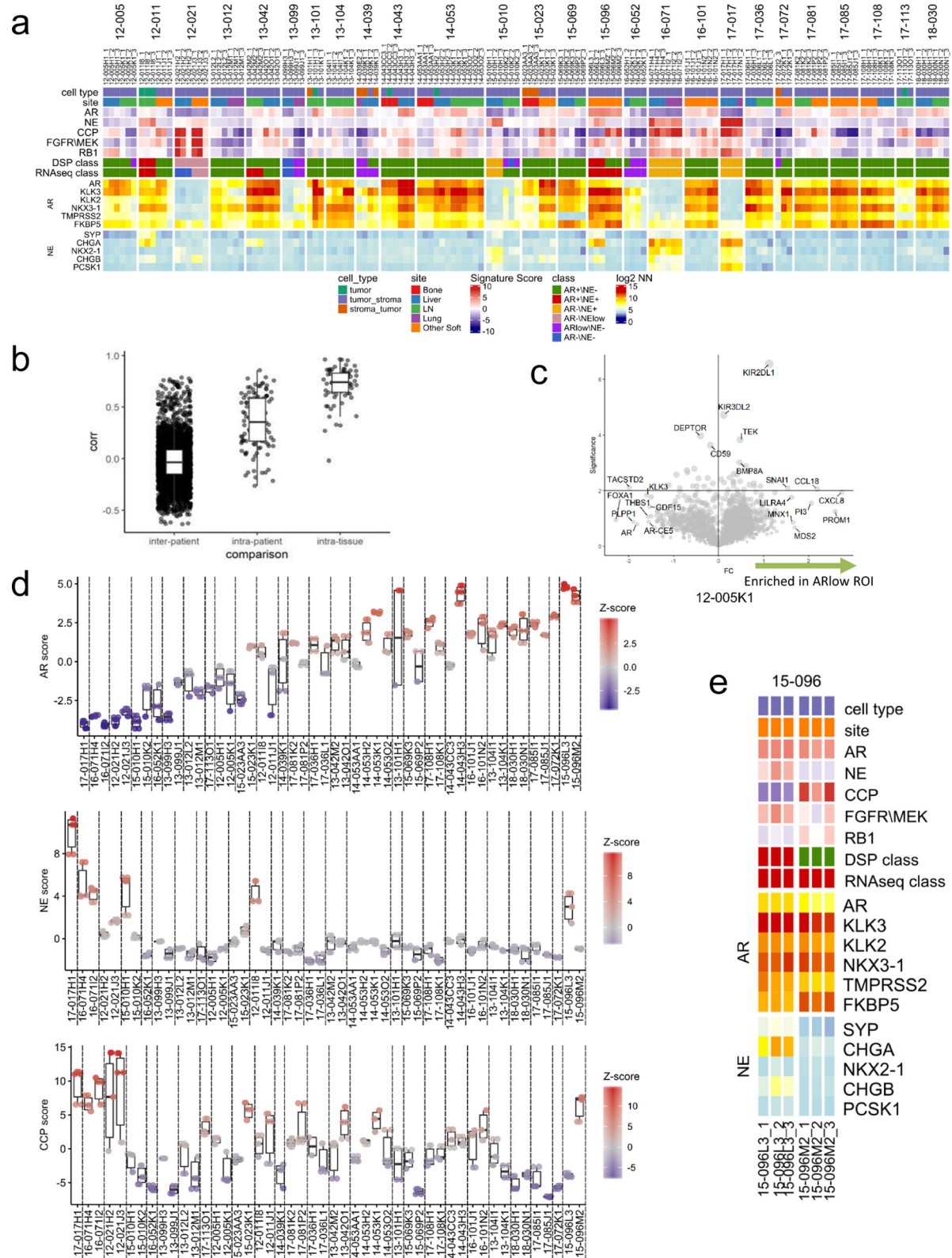

tumor diagnosis, provide insights into treatment resistance, and furnish biological rationale for new treatment strategies[50–52].

In this study, we used a robust approach for quantitative spatial molecular profiling to assess the inter- and intratumor variation in gene and protein expression of metastatic PCs. Key attributes of the DSP technology include the ability to quantitatively assess transcript and protein levels from standard FFPE biospecimens and the ability to sample multiple user-defined regions of interest specifically focused on defining heterogeneity and TME interactions. DSP accurately classified mPCs into subtypes such as ARPC and SCNPC, and assayed several biomarkers that are currently associated with specific therapeutics such as

**Fig. 4 Intratumoral gene expression homogeneity and heterogeneity. a** Heatmap of 141 ROIs from 53 individual tumor cores grouped by 26 patients highlighting androgen receptor (AR), neuroendocrine (NE), cell cycle progression (CCP), FGFR/MEK, and RB1 gene signatures. Results are expressed as gene signature Z-scores and presented according to color scale. **b** Boxplot demonstrating interpatient, intrapatient, and intratissue correlation across 141 individual DSP ROIs from 53 tumors from 26 patients. **c** Volcano plot indicating intrapatient heterogeneity in sample 12-005K1. The green arrow highlights genes enriched in AR^low^/NE− tumor core relative to the other two cores from the same tissue. **d** Data graphed as boxplots indicating differences in AR, NE, and CCP gene signature Z-scores across 141 ROIs from 26 patients included on the TMAs. Dotted lines separate each patient. The NE and CCP plots retain the patient ordering by the AR score as shown in the AR score plot. **e** Heatmap of six ROIs from two individual tumor cores (L3 and M2) from patient 15-096. Results are expressed as gene signature Z-scores and log$_2$ negative-normalized (NN) gene expression and presented according to the color scales in Fig. 2a. Exact intratumoral homogeneity was 40% based on the associated hypergeometric distribution for possible pairs of samples. Boxes in **b** and **d** represent the median and interquartile range (IQR) and the upper and lower whiskers extending to the values that are within 1.5 × IQR; data beyond the end of the whiskers are outliers and plotted as points.

PSMA/FOLH1. Though this study was not explicitly designed to evaluate assay performance as a function of biospecimen age, the FFPE samples spanning an 8-year time interval showed no age-related variation indicating that the platform is suitable both for retrospective studies as well as the analysis of biospecimens acquired in real time.

Prior studies have reported that mPCs within an individual share a common monoclonal origin and exhibit limited inter-tumoral heterogeneity with respect to driver mutations[10,18]. However, following AR-directed therapy, divergent mechanisms can contribute to resistance that include convergent evolution involving various alterations in the AR itself such as mutation, copy gains, and the expression of AR splice variants, as well as transdifferentiation to phenotypes that no longer rely on AR activity[11,19,21,53]. Through the analysis of the spatial composition of 53 metastases, we found common agreement in gene expression and phenotype classification between metastases from the same individual. Further, gene expression programs of spatially distinct intratumoral regions were also highly concordant. However, there were clear exceptions to this general conclusion as evidenced by the juxtaposition of AR+/NE− and AR−/NE+ tumor phenotypes within the same metastasis. The recognition that a tumor classified as amphicrine—AR+/NE+—by bulk tumor analysis, actually consisted of distinct subtypes, was readily demonstrated by DSP. This type of intratumoral heterogeneity has clear implications for therapy resistance. Furthermore, we identified intratumoral AR-V7 variant heterogeneity that generally correlated with IHC expression. There were a number of discordant cases present (DSP vs bulk tumor RNAseq), likely due in part to the heterogenous nature of AR-V7 expression in metastasis that has been described previously[54]. The discordant cases presented in this dataset reflect the need for further studies examining the capability of DSP in detecting splice variants, when compared to traditional approaches and determining if intratumoral variation in AR-V7 associates with treatment responses to ARSI therapy.

In addition to evaluating the attributes of neoplastic cells within the tumor mass, we used DSP to quantitate the intra-tumoral immune cell composition. A notable finding was the general lack of any substantial leukocyte population within PC metastases. Based on the assessment of both transcript and protein markers, macrophages constituted the most abundant immune cell type, but these were also generally uncommon. The low-to-absent expression of immune checkpoint proteins CTLA4, PD1, and PD-L1 is also congruent with the very low response rates of mPC patients to immune checkpoint blockade, excepting those with DNA mismatch repair deficiency and hypermutation[55–57]. A prior study of PC metastasis determined that tumors with deficient MMR expressed higher levels of several immune checkpoint proteins and harbored increased T-cell infiltrates[41]. None of the patients in the present study were MMR deficient.

In contrast to low/absent expression of immune checkpoint proteins for which there are FDA-approved antibodies, two immune checkpoint proteins, CD276/B7-H3 and TIM-3, were expressed at high levels across the majority of metastasis. Further, DSP demonstrated highly consistent expression of both CD276/B7-H3 and TIM-3 across the multiple ROIs within each tumor, indicating low intratumoral heterogeneity. Several functions have been attributed to B7-H3 including the inhibition of antitumor T-cell activity[58–60]. Notably, high B7-H3 expression in localized PC is associated with adverse outcomes following primary therapy[61,62]. A previous study evaluating B7-H3 expression by immunohistochemistry reported that 31 of 34 (91%) PC bone metastasis expressed moderate-to-high staining[63]. Our findings confirm this result and also demonstrate B7-H3 expression in metastases to other organs. Antibodies targeting B7-H3 have shown a favorable clinical safety profile and are currently in clinical trials for several solid tumors (NCT01391143, NCT04129320). Preclinical studies of CAR-T cells engineered to recognize B7-H3 demonstrate strong antitumor responses with very limited toxicity[64]. Conflicting reports exist on the role of TIM-3 in PC. Increased levels of TIM-3 were detected on CD4+ and CD8+ cells in patients with PC when compared to patients with benign prostatic hyperplasia[65]. Conversely, low protein expression of TIM-3 was associated with poor prognosis in patients with mPC, and identified as an independent predictor of CRPC[66]. While clarification of the role of TIM-3 in subtypes of mCRPC is required, the expression levels of B7-H3 and TIM-3 observed in the current study suggest the capability of DSP to identify potential therapeutic targets and support further studies of these immune modulatory proteins as therapeutic targets in mPC.

In conclusion, the present study focused on intratumoral assessments of gene expression and cell phenotype identification across and within metastatic tumors. In addition to delineating a high degree of concordance in the intratumoral phenotypic composition, we found a general lack of immune cell infiltrates in the vast majority of metastases and high expression of the immune checkpoint proteins B7-H3 and TIM-3. However, important contributors to tumor pathobiology reside at the tumor–host interface including immune cell components and paracrine-acting factors derived from cancer-associated fibro-blasts and other microenvironment cell types. Future studies employing high-plex DSP focused on interactions that occur in spatially restricted domains may identify additional mechanisms contributing to the lack of responses to immune-based therapy observed in patients with metastatic PC.

## Methods

**Study population.** Samples were obtained from patients who died of metastatic castration resistant PC and who had provided written informed consent as per the aegis of the Prostate Cancer Donor Program at the University of Washington. The Institutional Review Board of the University of Washington approved this study.

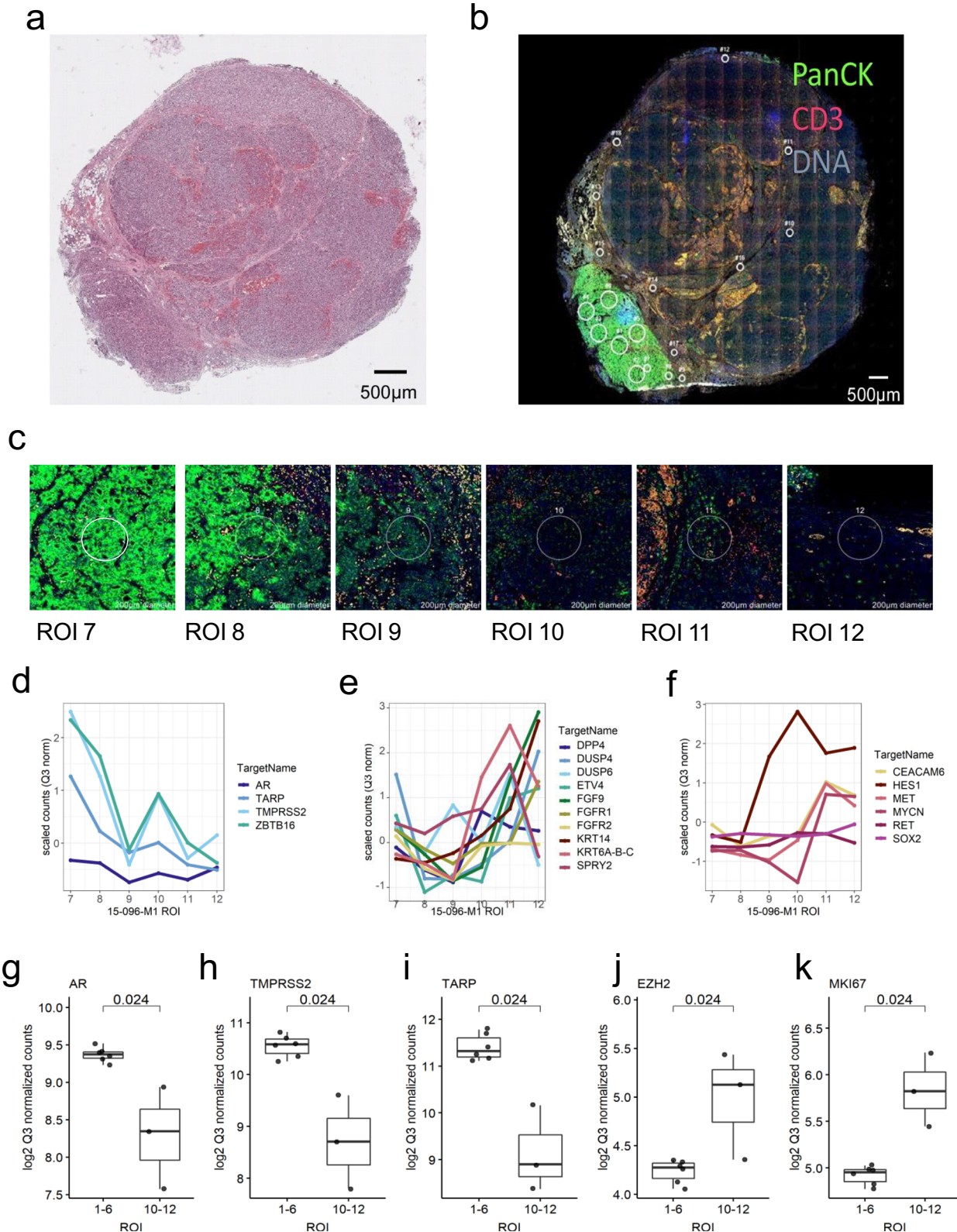

## TMA construction

The study design comprised specimens from 27 patients with two metastatic sites included per patient: 52 soft tissues metastases (liver $N = 17$, lymph node $N = 16$, lung $N = 6$, other $N = 13$) and four bone metastases. Visceral metastases were identified at the gross level, bone biopsies were obtained according to a previously described template from 16 to 20 different sites[67], and all metastases were verified at a histological level. Tissues were FFPE (bone metastases were decalcified in 10% formic acid). TMAs were constructed from three 1-mm diameter cores punched from each FFPE tissue block for a total of 168 tumor cores arrayed across three recipient TMA blocks.

## Immunohistochemistry

Five-micron thick sections of the TMAs were deparaffinized and rehydrated in sequential xylene and graded ethanol. Antigen retrieval was performed in 10 mM citrate buffer (pH 6.0) in a pressure cooker for 30 min. Endogenous peroxidase and avidin/biotin were blocked respectively (Vector Laboratories Inc.). Sections were then blocked with 5% normal goat–horse–chicken serum, incubated with primary antibody Anti-Androgen Receptor (Biogenex) MU256-UC (1:60), Anti-Androgen Receptor V7 antigen (clone RM7) (RevMab Biosciences) (1:2000), Anti-Prostate-Specific Antigen (Dako) A0562 1:1000, Anti-Synaptophysin (Santa Cruz) sc-17750 (1:200), incubated with biotinylated

**Fig. 5 Intratumoral heterogeneity within full-tumor section 15-096M1. a** Hematoxylin and eosin (H&E) staining of 15-096M1 lymph node metastases. Distinct areas of morphology are demonstrated, cribriform well differentiated prostatic adenocarcinoma (lower left) and undifferentiated high-grade carcinoma (upper right). $N = 1$ tissue section for H&E staining. **b** Fluorescent labeling of 15-096M1 lymph node metastases. High PanCK staining is present in the lower left and low PanCK staining is present in the upper right. $N = 1$ tissue section for fluorescent labeling. **c** Individual tumor region of interest (ROIs) (200–500 μm) with varying levels of PanCK intensity and differential tumor morphology. $N = 1$ tissue section for fluorescent labeling and ROI selection. Expression plots of genes known to be associated with AR+/NE− (**d**), AR−/NE− (**e**), and AR−/NE+ (**f**) phenotypes of ROIs 7–12 from 15-096M1. Counts were Q3 normalized and scaled (Z-score) to enable plotting of all genes on the same axes. **g–k** Comparison of transcript levels of specific genes in ROIs 1–6 ($N = 6$) from the CK+ tumor region and ROIs 10–12 ($N = 3$) distant from the CK+ region. Counts were log$_2$ Q3 normalized. Significance was determined by two-sided Wilcoxon-rank tests (**g–k**: $p = 0.024$). Boxes represent the median and interquartile range (IQR) and the upper and lower whiskers extending to the values that are within 1.5 × IQR; data beyond the end of the whiskers are outliers and plotted as points.

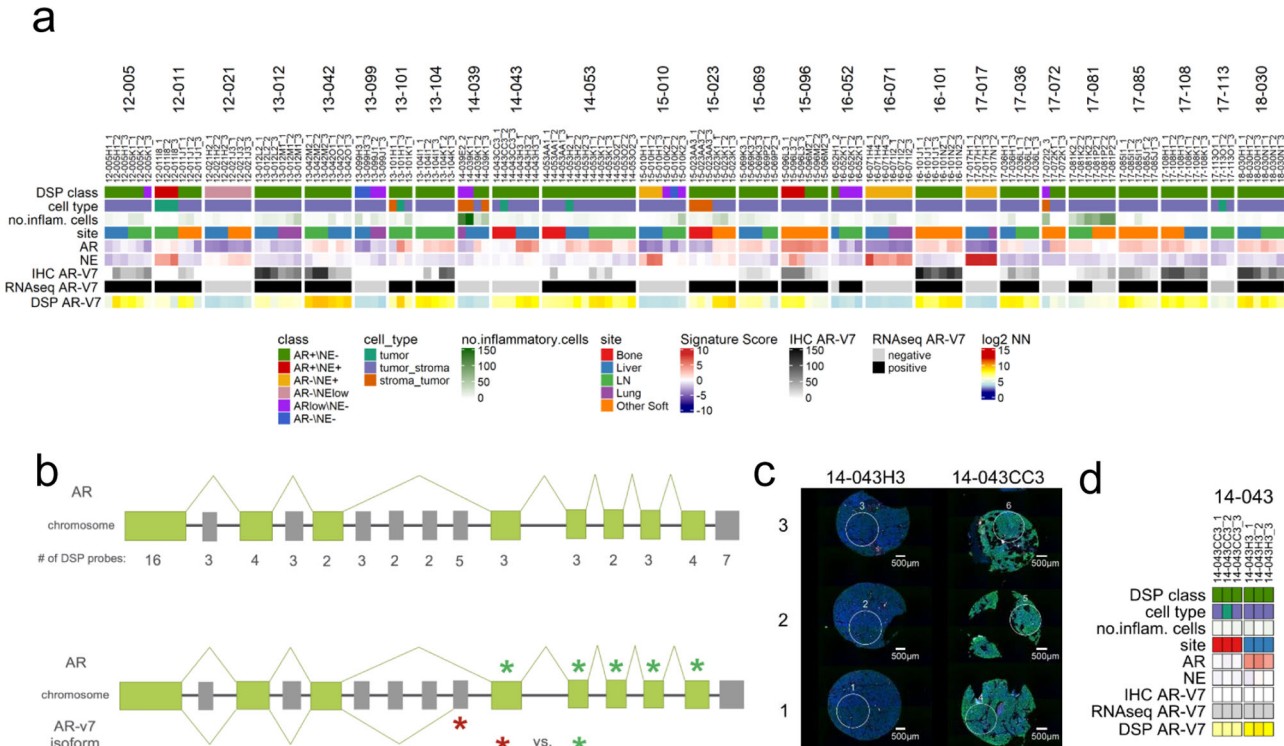

**Fig. 6 Alternative splicing of AR isoforms is present across metastases, as determined by DSP. a** Heatmap of 141 individual tumor ROIs grouped by 26 patients comparing bulk RNAseq AR-V7 expression to DSP. Results are expressed as gene signature Z-scores and log$_2$ negative-normalized (NN) gene expression and presented according to color scales. **b** DSP–RNA probe design of AR full length and AR-V7 variant isoform. **c** Fluorescent labeling and ROI selection of three cores from two sites of metastasis from patients 14-043 that demonstrated divergent AR-V7 expression determined by RNAseq and DSP. $N = 1$ TMA section for fluorescent labeling. **d** Heatmap demonstrating discordance in AR-V7 expression DSP AR-V7 and AR-V7 measured by bulk RNAseq and AR-V7 immunohistochemistry (IHC) in six tumor ROIs from two sites (CC3 and H3) within patient 14-043. Results are presented according to color scales in **a**.

secondary antibody (Vector Laboratories Inc.), followed by ABC reagent (Vector Laboratories Inc.) and stable DAB (Invitrogen Corp.). All sections were lightly counterstained with hematoxylin and mounted with Cytoseal XYL (Richard Allan Scientific). Mouse (MOPC-21 developed by the Genitourinary Cancer Research Lab at the University of Washington) or rabbit (I-1000-5 Vector Labs) IgG were used as negative controls at the same concentration as the primary antibodies. IHC staining was evaluated by a pathologist (M.R.) using a scoring system created by multiplying three intensity staining levels (0, no staining; 1, weak staining; and 2, strong staining) by the percentage of cells at each staining level and summing the two values. The sum provided a final score for each sample (score range was 0–200). The final score for each sample was the average of the scores of each triplicate or the average value of two, if one was missing[68].

**RNAseq analysis.** Total RNA was isolated from 56 metastases fresh frozen in OCT (Tissue-Tek) with RNA STAT-60 (Tel-Test) and extracted by column purification using an RNA isolation kit RNeasy (Qiagen) according to the manufacturer's protocol. RNA was quantified by NanoDrop2000 spectrophotometer (Thermo Fisher) and integrity determined on a 2100 Bioanalyzer (Agilent Technologies). RNAseq libraries were constructed from 1 μg total RNA using the Illumina TruSeq Stranded mRNA LT Sample Prep Kit according to the manufacturer's protocol. Barcoded libraries were pooled and sequenced on the Illumina HiSeq 2500

generating 50 bp paired end reads. Sequencing reads were mapped to the hg38 human using STAR v2.7.3a[69]. Gene level abundance was quantitated from the filtered human alignments in R using the GenomicAlignments Bioconductor package (version 1.22.1)[70]. AR-V7 was quantified by summing gapped reads within the exon 3 to CE3 junction (chrX:67686127-67694672) from the aligned BAM files using the GenomicAlignments Bioconductor package (version 1.22.1) and normalized as the spliced reads per million[12]. The RNAseq data used in this study are available under GEO accession number GSE147250.

**In situ hybridization.** To prepare slides for DSP, 4-μm thick TMA sections were deparaffinized, heated in ER2 solution (Leica) at 100 °C for 20 min, and treated with 1 μg/ml proteinase K (Ambion) at 37 °C for 15 min on a BOND RXm autostainer (Leica). An overnight in situ hybridization was performed as described[71] with a final probe concentration of 4 nM per probe. The panel included probes that target 2106 mRNA transcripts as well as 220 negative probes (18,120 probes total, median 10 probes per target). Slides were washed twice at 37 °C for 25 min with 50% formamide/2X SSC buffer to remove unbound probes.

**Sample preparation and analysis for multiplexed protein profiling with GeoMx.** Slides were deparaffinized and rehydrated in staining jars by incubating for 3 × 5 min in CitriSolv, 2 × 5 min in 100% ethanol, 2 × 5 min in 95% ethanol, and

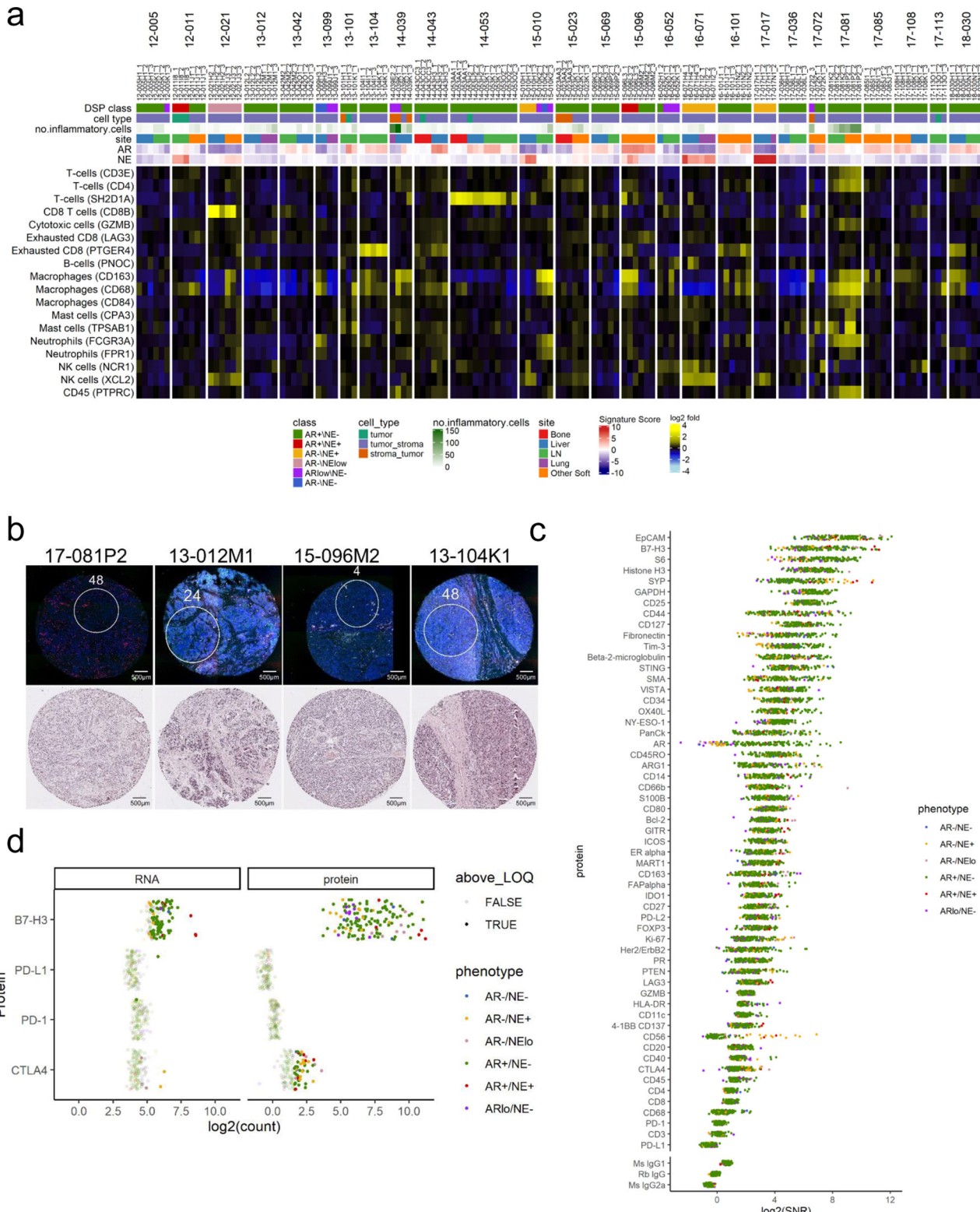

**Fig. 7 DSP describes immune cell microenvironments of distinct phenotypes of mCRPC. a** Heatmap of DSP immune signaling genes across 141 individual regions of interest (ROIs) from 26 patients. Results are expressed as gene signature $Z$-scores and $\log_2$ mean-centered gene expression and presented according to color scales. **b** Fluorescently labeled patient core with matched hematoxylin and eosin (H&E) staining representing high and low levels of inflammatory infiltrate. High—17-081P2 is comprised of 70% tumor, 30% stroma, with 100 CD3+ leukocytes present, and 13-012M2 is comprised of 80% tumor, 20% stroma, with 40 CD3+ leukocytes present. Low—15-096M2 is comprised of 90% tumor, 10% stroma with zero CD3+ cells, and 13-104K2 is comprised of 90% tumor, 10% stroma with three CD3+ cells present. Immune cells counted based on CD3 immunohistochemical staining. $N = 1$ TMA section for fluorescent labeling and H&E staining. **c** DSP protein depicts overall low levels of intratumoral immune cells. Data are graphed as $\log_2$ signal-to-noise ratio (SNR). **d** Consistently high expression of B7-H3 is present in the ARpos_NEneg phenotype when compared to other CRPC phenotypes. Expression is consistent across RNA and protein DSP in B7-H3, PD-L1, PD1, with slightly higher expression in protein DSP observed for CTLA4. Data are presented as $\log_2$ normalized counts.

2 × 5 min in deionized water. Antigen retrieval was performed by placing a staining jar containing the slides and 1X Citrate Buffer (pH 6) into a pressure cooker at high temperature and high pressure for 15 min. After releasing the pressure, the staining jar was left at room temperature with its lid removed for 25 min. Slides were then washed 5 × 1 min in 1X TBS-T. Blocking was performed by placing slides horizontally in a humidity chamber and covering the tissue with Buffer W (NanoString) for 1 h at room temperature. A mixture of the detection antibodies and morphology markers was diluted in Buffer W to a final concentration of ~0.25 μg/ml for each antibody. The antibody panel consisted of 60 oligo-conjugated antibodies described previously[71] with the addition of antibodies recognizing AR and SYP (Supplementary Data File 5: Note, NanoString provides a NanoString Protein Probe ID for each unique antibody conjugation clone in lieu of specific identifying information about the antibodies as NanoString considers this confidential information). Each antibody was assigned a single probe ID. After removing the blocking solution, the diluted antibody mixture was pipetted onto the slides and the humidity chamber was incubated at 4 °C overnight. Slides were washed 3 × 10 min in 1X TBS-T and then postfixed in 4% PFA for 30 min at room temperature, followed by 2 × 5 min washes in 1X TBS-T. Nuclei were stained with 500 nM SYTO 13 for 15 min at room temperature in a humidity chamber and rinsed with 1X TBS-T before loading onto the GeoMx instrument.

**Digital spatial profiling.** Prepared slides were stained with immunofluorescent antibodies to facilitate the identification of tissue morphology: pan-cytokeratin (AE1 + AE3, Novus Biologicals) for epithelial cells and CD3 (UMAB54, OriGene labeled with Alexi Fluor 647) or CD45 (2B11 + PD7/26, Novus Biologicals) for T cells, as well as the DNA stainSyto13 (NanoString, 121303303). Stained slides were loaded onto a GeoMx instrument and scanned. One circular ROI measuring 500 μm in diameter was selected per tissue core. Each ROI was annotated post selection by a pathologist (M.R.) to estimate tumor and stromal content (%). ROI locations for RNA and protein slides were selected to be the same wherever possible. GeoMx DSP technology is for research use only and not for use in diagnostic procedures.

**Library preparation and sequencing.** Collected oligos from each ROI were PCR amplified using a forward primer with the sequence CAAGCAGAAGACGGCAT ACGAGATXXXXXXXXXGTGACTGGAGTTCAGACGTGTGCTCTTCCGATCT and a reverse primer with the sequence AATGATACGGCGACCACCGAGAT CTACACXXXXXXXXXXACACTCTTTCCCTACACGACGCTCTTCCGATCT, where Xs represent custom Illumina i5/i7 unique dual indexing sequences to preserve ROI identity. PCR products were pooled and purified twice with AMPure XP beads (Beckman Coulter). PCR products for each analyte type (RNA and protein) were pooled and sequenced separately. Library concentration and purity were measured using a high sensitivity DNA Bioanalyzer chip (Agilent). Paired end (2 × 38 bp reads) sequencing was performed on an Illumina NextSeq instrument.

**Data processing and analysis.** After sequencing, reads were trimmed, merged, and aligned to retrieve the probe identity. The unique molecular identifier region of each read was used to remove PCR duplicates and duplicate reads, thus converting reads into digital counts. The sequencing saturation was sufficient for both RNA and protein analytes at 63 and 82%, respectively. For each gene in each sample, the reported count value is the mean of the individual probe counts after removal of outlier probes.

The LOQ was set at the geometric mean plus two standard deviations of the negative probes. Of 2093 genes targeted by DSP, 1636 (78%) were above the detection threshold in at least one ROI. The 457 genes below the detection threshold in all ROIs were excluded from further analysis. Enrichment scores were calculated in R using the Z-scores function within the GSVA package (version 1.32.0)[72] with default parameters and $\log_2$ negative-normalized expression values above background as input. All signatures used are described in Nyquist et al.[29]. Sample phenotypic groups were visualized using classical MDS calculated with the cmdscale function in R using the expression of 23 out of 26 genes in a published gene signature[23]. Three genes (ACTL6B, S100A14, and FGFBP1) were removed due to lack of expression in the DSP dataset. The distance metric was "Euclidean" calculated by dist function on the columns (samples). Pearson's correlation coefficient was used to study the relationships between variables shown in scatterplots using the cor.test function in R. The counts for each antibody were divided by the geometric mean of the three IgG negative control antibodies on a per ROI basis to create a signal-to-noise ratio (SNR). Antibodies below a SNR of 3 are considered not detected in that ROI. Protein signal between ROIs was normalized using three positive reference or housekeeping proteins (ribosomal protein S6, histone H3, and GAPDH). Homogeneity of samples from the same tumor or patient was quantified using the mean proportion of randomly sampled pairs of samples that agree in terms of nonmissing phenotype classification (six categories); uncertainty was quantified using bias-corrected and accelerated 95% confidence intervals based on 1000 bootstrap replicates. All analyses utilizing R were performed with version 3.5.1 or 3.6.2 and RStudio 1.3.1093.

**Reporting summary.** Further information on research design is available in the Nature Research Reporting Summary linked to this article.

## Data availability

All of the relevant data for this study are publicly available and have been provided by the authors. The RNAseq data used in this study are available under GEO accession number GSE147250. The DSP transcript data are provided in Supplementary Data File 3. The DSP protein data are provided in Supplementary Data File 4. The remaining data are available within the article, Supplementary Information, or available from the authors upon request.

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

## Acknowledgements

The authors would like to thank the patients and their families who participated in these studies. The authors would like to thank Celestia Higano, Evan Yu, Heather Cheng, Mike Schweizer, Bruce Montgomery, Elahe Mostaghel, Daniel Lin, Funda Vakar-Lopez, Eva Corey, and the rapid autopsy teams for their contributions to the University of Washington Medical Center Prostate Cancer Donor Rapid Autopsy Program. The authors would like to thank members of the Nelson, Haffner, Vasioukhin, and Lee laboratories for constructive suggestions. The authors gratefully acknowledge research support from Cancer Center Support Grant P30CA015704-40, NIH awards P50CA97186, R01CA234715, P01CA163227, U54 CA224079, R50CA221836, and S10OD028685, CDMRP Awards W81XWH-18-1-0406, W81XWH-18-1-0347, and W81XWH-18-1-0354, the Prostate Cancer Foundation, the Institute for Prostate Cancer Research, Veterans Affairs Research Service (SRP), and the Department of Defense, Prostate Cancer Biorepository Network (W81XWH-14-2-0183).

## Author contributions

L.B. and P.S.N. designed and supervised the research. C.M. provided the biospecimens. M.H., G.G., and J.B. supervised the DSP experiments and provided support. Z.Z. and M.H. prepared DSP–RNA reagents and performed DSP–RNA experiments. B.B. and R.M. prepared DSP protein reagents and performed DSP protein experiments. L.B. and C.M. selected DSP regions of interest for molecular profiling. S.R.P. performed AR-V7 immunohistochemistry analysis. L.D.T. and M.R. evaluated the histology and immunohistochemistry. M.K. and I.C. analyzed the DSP and RNAseq data. R.G. performed statistical analysis. L.B., M.K., and P.S.N. wrote the manuscript. All authors reviewed and edited the final manuscript.

## Competing interests

P.S.N. received instrument support (GeoMx) from NanoString Technologies and consulting fees from Astellas, Janssen, and Bristol Myers Squibb for services unrelated to the present work. M.K., Z.Z., B.B., R.M., G.G., M.H., and J.B. are employees and stockholders of NanoString Technologies. The authors received appropriate permissions for use of NanoString Technologies figures depicting DSP instrument workflow. L.B., C.M., I.C., M.R., L.D.T., R.G., and S.R.P. declare no competing financial interests in relation to this work.
