## [Peer Review File · Nature Communications]

REVIEWER COMMENTS

Reviewer #1 (Remarks to the Author): Expert in prostate cancer genomics

The manuscript describes the application of a relatively recent technology (Nanostring DSP) for spatial gene and protein expression profiling of metastatic prostate cancer tissue samples. The spatial aspect as well as the association of gene and protein expression contribute a substantial degree of novelty to the literature. Overall, the report consists of a set of well-performed experiments, which are of high interest for readers working in the fields of prostate cancer research/therapy and tumor heterogeneity.

Minor comments:

p. 6, line 124/5: It is somewhat counterintuitive to focus on intra-tumor heterogeneity using 3 cores per ROI and afterwards average (why average rather than median?) the three values. Variability is expected to be high in those cases, but the individual values should be displayed (e.g. in a supplemental table).

It would be helpful to learn more about the definition of DSP classes (figures 2,4) without having to extract this information from the previous paper by the same group. Many of the arguments in the present report build on this definition. Therefore, a brief explanation would be appreciated. Were quantitative expression thresholds of AR and NE (which markers?) used? If so, please indicate the values.

The molecular homogeneity between metastases from different sites in the same patients is remarkable and deserves more investigation w.r.t. their clinical relevance. However, the description of this aspect deserves more detail. This concerns, e.g., the finding of high concordance in most of the sample pairs (p. 7) but also the enumeration and a more elaborate description of the 7 discordant cases. It should be made clear that Figure 3d just represents a section of 3a etc.

In a similar logic, the reader should be better guided through the concordant and discordant cases of intra-tumoral homo- and heterogeneity (p. 8/9 and associated figure 4d).

A reference to the corresponding ethics vote of the commission at the University of Washington should be given.

Reviewer #2 (Remarks to the Author): Expert in spatial transcriptomics

In this manuscript, Brady et al. present a comprehensive dataset of human metastatic prostate tumors studied using the Nanostring technology. The dataset appears to be highly interesting as it is described. However, the data does not seem to be exploited to its full potential and no major claims are made.

The relationship between patients, sites, samples, cores, ROIs could be clarified with a schematic. Not all of the 56 metastases were used (we counted 46 sites in figure 4a) and this should be indicated also in the Figure 1a schematic.

The authors claim (in the abstract) “we found a high level of homogeneity with respect to tumor

phenotype” It is unclear what this means, homogeneity within a patient? Within a sample? What are the criteria for homogeneity? This needs a statistical model. Figure 4e shows a heatmap for 2 metastases, but the degree of homogeneity could be quantified, and this analysis extended to the other samples. Additionally, figure 4e has information (first 9 rows; half of the subfigure) that is completely redundant with a part of Figure 1a. Also why are the authors not showing us the other information in 4e (the non-redundant part) for the other patients.

The authors also describe an exception to this homogeneity, with “high and low androgen receptor (AR) and neuroendocrine activity” but they do not describe the corresponding Figure 4c analysis in any specific detail in the results section. Why is that particular patient shown? What is being concluded? They mention three tumors (lin 213) but the analysis in 4c is only shown for one tumor. Also the genes up-regulated in ARlow cores should be compared across patients. How many of the genes identified are part of the AR signature? These should serve as a positive control, and do not provide new insights.

The authors should re-order the patients in Figure 4d, according to AR score for example (or NE, or CCP scores). This may reveal patterns relating to the other scores, and would generally be more informative.

The authors claim (in the abstract) that “B7-H3/CD276 immune checkpoint protein was highly expressed, particularly in mPCs with high AR activity”. However, this has been previously published (“B7-H3 and B7x are highly expressed in human prostate cancer and associated with disease spread and poor outcome; <https://pubmed.ncbi.nlm.nih.gov/18042703/>). It is encouraging that the present dataset supports this result however, the original manuscript should be cited and this should be described as an original result.

The conclusions of the figure 5 analysis are not supported by statistics.

The authors compare their method to bulk RNA-Seq to highlight its use. However, detecting heterogeneity within a tumor is possible using other methods including single-cell RNA-Seq and spatial transcriptomics. The authors should note this. However, most RNA-Seq based methods do not distinguish between splicing isoforms, whereas this method does. This is the basis of the figure 6 analysis, but this strength could be further highlighted and exploited.

Finally, half of the authors are NanoString employees and Figure 1b is adapted from material used for marketing the Nanostring platform (<http://enseqlopedia.com/2018/05/nanostring-in-cambridge/>). In the very least, this should be acknowledged as a competing financial interest.

Reviewer #3 (Remarks to the Author): Expert in spatial proteomics

In the present study by Brady et al entitled: “Inter- and Intra-tumor Heterogeneity of Metastatic Prostate Cancer Determined by Digital Spatial Gene Expression Profiling”, the authors assessed multiple discrete areas across multiple prostate cancer metastases and found a high level of

homogeneity with respect to tumor phenotype. In addition, the authors observed exceptions in tumors comprising regions with high and low androgen receptor (AR) and neuroendocrine activity. Authors' results demonstrate the utility of rather new technique digital spatial profiling (DSP) for accurately classifying tumor phenotype, assessing tumor heterogeneity, and identifying new aspects of tumor biology involving the immunological composition of metastases.

While the work is of merit and results are really sound some concerns avoid its recommendation as it is.

Concerns:

It would be recommended since the correlation at transcript and protein level was observed regarding to 2093 genes but about 60 proteins, to increase by other means the correlation between RNA and protein (i.e. immunohistochemistry, WB, qPCR) from selected targets.

The manuscript would be benefitted if the authors will be able to discuss potential therapeutic targets analyzed by DSP and found to be dysregulated (or not) and could aid in the therapy of patients with no response to immune-based therapy to provide further information of the usefulness of DSP to provide data to manage patients.

Regarding ARv7 expression in those tumors with discordance between Arv7 status measured by DSP and RNAseq, have the authors considered to verify by other means (i.e. amplification of Arv7 transcript by PCR) whether DSP or RNAseq status fail?. Depending on the results, this will reinforce the usefulness of DSP to detect alternative splicings.

RE: # NCOMMS-20-27107
Nature Communications
Editorial Staff and Reviewers

REVIEWER COMMENTS – Point by point response to manuscript submission # NCOMMS-20-27107

We thank the reviewers for their thorough assessment of our manuscript and constructive comments and suggestions. Below is a point-by-point response and notations for changes/updates to the manuscript results.

Author note: Since the original submission of this manuscript, we have completed next generation sequencing readout data for the DSP protein experiments (previously n-counter). The methods section has been updated and the following figures and corresponding captions have been replaced with the NGS readout, in the same figure format as the original submission, throughout the manuscript: Figure 1b, 1e, 1f, 7c, 7d and Figure S3c,S3d, and S3e. The modified analysis approach has not altered our findings.

Reviewer #1 (Remarks to the Author): Expert in prostate cancer genomics

The manuscript describes the application of a relatively recent technology (Nanostring DSP) for spatial gene and protein expression profiling of metastatic prostate cancer tissue samples. The spatial aspect as well as the association of gene and protein expression contribute a substantial degree of novelty to the literature. Overall, the report consists of a set of well-performed experiments, which are of high interest for readers working in the fields of prostate cancer research/therapy and tumor heterogeneity.

Minor comments to address:

Comment 1: *p. 6, line 124/5: It is somewhat counterintuitive to focus on intra-tumor heterogeneity using 3 cores per ROI and afterwards average (why average rather than median?) the three values. Variability is expected to be high in those cases, but the individual values should be displayed (e.g. in a supplemental table).*

Response: Thank you for your comments and review of this manuscript. We have included the individual values for each measurement for each ROI in Supplementary Table 2 columns R-V. Each column denotes Z-values for individual ROIs for each of the AR, NE, CCP, FGF/MEK and RB1-loss activity scores. Mean values were only used in order to compare DSP to bulk RNA-seq data as we feel that the mean is a better approximation to a bulk mixture.

Comment 2: *It would be helpful to learn more about the definition of DSP classes (figures 2,4) without having to extract this information from the previous paper by the same group. Many of the arguments in the present report build on this definition. Therefore, a brief explanation would be appreciated.*

Response: The manuscript has been updated to include a description of each DSP class (page 6; starting with line 129) within the main body of the manuscript.

Comment 3: *Were quantitative expression thresholds of AR and NE (which markers?) used? If so, please indicate the values.*

Response: Quantitative expression thresholds of AR and NE signatures were not used. Other than applying expression thresholds globally to the dataset (see methods lines 487-492), the phenotypes are assigned by their position in the MDS plot of 23 genes rather than by individual AR and NE gene expression criteria.

Comment 4: *The molecular homogeneity between metastases from different sites in the same patients is remarkable and deserves more investigation w.r.t. their clinical relevance. However, the description of this aspect deserves more detail. This concerns, e.g., the finding of high concordance in most of the sample pairs (p. 7) but also the enumeration and a more elaborate description of the 7 discordant cases.*

Response: A non-parametric statistical model was added to the manuscript (pages 7 and 17) to quantify both intra-patient and intra-tumoral homogeneity of phenotype based on AR and NE activity. Further, Figure 3d has been amended to include all discordant cases to provide additional clarity. We concur that homogeneity/heterogeneity may be an important factor in terms of clinical relevance regarding tumor progression and particularly response to treatment. Future studies will incorporate DSP approaches to assess tumor diversity with respect to outcomes.

Comment 5: *It should be made clear that Figure 3d just represents a section of 3a etc.*

Response: The caption for Figure 3d has been revised to reflect this adaptation (page 25).

Comment 6: *In a similar logic, the reader should be better guided through the concordant and discordant cases of intra-tumoral homo- and heterogeneity (p. 8/9 and associated figure 4d).*

Response: As noted in the response to Comment 4, a statistical model has been added to the manuscript to summarize the degree of homogeneity observed across patients and across tumors from the same patient.

Figure 4d has been updated to reorder patients by signature score, following the same color scheme as utilized in figure 4a. We hope this improves the orientation for the reader in understanding intra- and inter-tumor homo- and heterogeneity.

Comment 7: *A reference to the corresponding ethics vote of the commission at the University of Washington should be given.*

Response: An ethics statement has been added to the end of the manuscript (page 17) detailing ethical approval of this study. The acknowledgements section has been further updated to thank involvement of the Rapid Autopsy Program team members.

Reviewer #2 (Remarks to the Author): Expert in spatial transcriptomics

In this manuscript, Brady et al. present a comprehensive dataset of human metastatic prostate tumors studied using the Nanostring technology. The dataset appears to be highly interesting as it is described. However, the data does not seem to be exploited to its full potential and no major claims are made.

Comment 1: *The relationship between patients, sites, samples, cores, ROIs could be clarified with a schematic. Not all of the 56 metastases were used (we counted 46 sites in figure 4a) and this should be indicated also in the Figure 1a schematic.*

Response: Thank you for your comments related to this manuscript. Figure 1a and the corresponding caption (page 24) have been updated to describe ROIs removed from the analysis. For clarification, in Figure 4a there are 53 sites and 141 ROIs. Different sites are denoted by the capital letter as part of the patient ID e.g. 12-005H1, H refers to the metastatic site. Table S2 provides detailed information for each ROI.

Comment 2.i *The authors claim (in the abstract) “we found a high level of homogeneity with respect to tumor phenotype” It is unclear what this means, homogeneity within a patient? Within a sample? What are the criteria for homogeneity? This needs a statistical model.*

Response: The high level of homogeneity referenced in the abstract refers to intra-patient homogeneity. This has been clarified (page 2). We also found a high level of intra-tumoral homogeneity. As implied by the reviewer, homogeneity and heterogeneity can be measured in many ways. In this study, we have focused on the phenotypic classification of a metastasis, as this has practical value with respect to allocating an AR pathway antagonist, and possibly allocating therapy for neuroendocrine/small cell cancer (e.g. platinum). Consequently, we assessed the homogeneity or heterogeneity of phenotype classification. A non-parametric statistical model was added to the manuscript (pages 7 and 17) to quantify homogeneity. The model estimates the probability that randomly sampled pairs of samples from the same patient have the same phenotype classification (i.e. 1 of 6 combinations of AR and NE activity) (Table S2). Based on this model, there is an 82% probability (95% CI 50%-92%) that randomly selected pairs of regions from the same patient would have the same phenotype classification.

Comment 2.ii *Figure 4e shows a heatmap for 2 metastases, but the degree of homogeneity could be quantified, and this analysis extended to the other samples.*

Response: The caption for Figure 4e has been amended (page 25) to include the degree of homogeneity (40%) based on an exact probability calculation consistent with the statistical model mentioned in the previous response.

Figure 4e has also been updated to amend our initial error which omitted an ROI. All three ROIs for both tumors are currently represented.

Comment 2.iii *Additionally, figure 4e has information (first 9 rows; half of the subfigure) that is completely redundant with a part of Figure 1a. Also why are the authors not showing us the other information in 4e (the non-redundant part) for the other patients.*

Response: We expanded the panel out of the larger Figure 1a in order to more clearly show the data and discussion of the results. Figure 4a has been revised to include the AR and NE genes for all patients. Figure 4e has been included to demonstrate/emphasize RNA seq vs. DSP discordance.

Comment 2.iv *The authors also describe an exception to this homogeneity, with “high and low androgen receptor (AR) and neuroendocrine activity” but they do not describe the corresponding Figure 4c analysis in any specific detail in the results section. Why is that particular patient shown? What is being concluded? They mention three tumors (line 213) but the analysis in 4c is only shown for one tumor.*

Response: We apologize for not providing more detail. The intent is to demonstrate that though homogeneity is the rule, there are clear outliers where heterogeneity with respect to tumor phenotype classification is evident. Our intent was not to focus on each deviation. We highlighted one patient and the derived tumor (12-005K1) simply as an example. Figure 4d shows inter- and intra-tumoral homogeneity/heterogeneity for additional parameters (e.g. cell cycle – CCP score). The manuscript has been updated on page 8, 9 and page 25 to reflect these queries and provide clarity on the relevance of Figure 4c. Of note, we also provide a statistical assessment of intratumoral homogeneity with respect to phenotype classification: “Overall, there was high intra-tumor concordance for the transcript-defined tumor phenotypes with a 96% probability (95% CI 79%-98%) that randomly sampled ROI pairs from the same tumor have the same phenotype classification”.

Comment 2.v *Also the genes up-regulated in ARlow cores should be compared across patients. How many of the genes identified are part of the AR signature? These should serve as a positive control, and do not provide new insights.*

Response: We apologize if the intent of this figure and results were not clear. The intent was to demonstrate intra-tumoral heterogeneity, as determined by phenotype classification and gene

expression when comparing ROIs from the same metastasis. This comparison demonstrated an AR^{low} ROI and an AR⁺ROI. We do not expect genes from the AR signature to be upregulated in the AR^{low} ROI, but rather downregulated (or expressed at a lower level) compared to the AR⁺ ROI. An example of this is KLK3 downregulated in the volcano plot as part of Figure 4c. We do not expect to see the same expression patterns in the multiple samples mentioned in the manuscript, as one (12-005K1) has tumors with both a AR^{low}/NE⁻ and AR⁺/NE⁻ phenotype, whereas (15-010K2) has tumors with both AR^{low}/NE⁻ and AR⁻/NE⁻ phenotypes. As the AR^{low}/NE⁻ ROIs were compared only to the corresponding ROIs within the same tumor, and different phenotypes exist, expression patterns may be dissimilar.

As part of this response, we compared both the AR^{low}/NE⁻ ROIs from 12-005K1 and 15-010K2 (A) and all AR^{low}/NE⁻ ROIs (B) to all AR⁺/NE⁻ ROIs in the dataset to determine downregulation of AR related genes in this phenotype. AR associated genes are downregulated in both analyses, included KLK2, KLK3, NKX3.1 and TARP.

These data are not presently included in the manuscript.

Comment 3: The authors should re-order the patients in Figure 4d, according to AR score for example (or NE, or CCP scores). This may reveal patterns relating to the other scores, and would generally be more informative.

Response: Figure 4d has been updated to reorder patients by signature score, following the same color scheme as utilized in figure 4a.

Comment 4: The authors claim (in the abstract) that “B7-H3/CD276 immune checkpoint protein was highly expressed, particularly in mPCs with high AR activity”. However, this has been previously published (“B7-H3 and B7x are highly expressed in human prostate cancer and associated with disease spread and poor outcome; <https://pubmed.ncbi.nlm.nih.gov/18042703/>). It is encouraging that the present dataset supports this result however, the original manuscript should be cited and this should be described as an original result.

Response: The manuscript has been updated to add this citation (page 13). In the original submission we included a citation to a prior study demonstrating B7-H3 expression in PC (Ref 64). We note that the study cited by the reviewer – Zang et al evaluated B7-H3 in primary prostate cancer and included no analysis of PC metastasis. As ICB is primarily used as a therapy for metastatic cancers, determining the status/expression of immune checkpoints in metastases is the most clinically relevant analysis. The present study provides such an assessment.

Comment 5: The conclusions of the figure 5 analysis are not supported by statistics.

Response: We thank the reviewer for this comment; however, the figure 5d-f analysis was intended to be qualitative due to the small number of samples and genes shown. We feel that a statistical analysis is not valid in this context and there are no claims about statistical significance in the text. Of note, Figures 5d, 5e and 5f have been updated to Q3 normalization. This has not altered the observations.

Comment 6: *The authors compare their method to bulk RNA-Seq to highlight its use. However, detecting heterogeneity within a tumor is possible using other methods including single-cell RNA-Seq and spatial transcriptomics. The authors should note this. However, most RNA-Seq based methods do not distinguish between splicing isoforms, whereas this method does. This is the basis of the figure 6 analysis, but this strength could be further highlighted and exploited.*

Response: We concur with the reviewer regarding other methods/approaches for assessing heterogeneity. The oldest approach is simply histology and immunohistochemistry. We have cited other methodologies on page 12 in the 'Discussion' section: *"New technologies such as spatial transcriptomics, multiplexed immunofluorescence, CODEX and mass cytometry are capable of integrating multiplexed molecular assays with a spatial context which can detect heterogeneity and may enhance the accuracy of tumor diagnosis, provide insights into treatment resistance, and furnish biological rationale for new treatment strategies⁵²⁻⁵⁴".*

For the revision, AR-V7 IHC was performed to further validate the DSP findings regarding splice variants. The manuscript has been amended on pages 10 and 13.

Comment 6: *Finally, half of the authors are NanoString employees and Figure 1b is adapted from material used for marketing the Nanostring platform (<http://enseglope-dia.com/2018/05/nanostring-in-cambridge/>). In the very least, this should be acknowledged as a competing financial interest.*

Response: We apologize for not including such a statement at the outset. The conflicts section on page 1 of the manuscript has been updated to include acknowledgments of instrument support, employee and stockholder involvement of named authors.

Reviewer #3 (Remarks to the Author): Expert in spatial proteomics

In the present study by Brady et al entitled: "Inter- and Intra-tumor Heterogeneity of Metastatic Prostate Cancer Determined by Digital Spatial Gene Expression Profiling", the authors assessed multiple discrete areas across multiple prostate cancer metastases and found a high level of homogeneity with respect to tumor phenotype. In addition, the authors observed exceptions in tumors comprising regions with high and low androgen receptor (AR) and neuroendocrine activity.

Authors' results demonstrate the utility of rather new technique digital spatial profiling (DSP) for accurately classifying tumor phenotype, assessing tumor heterogeneity, and identifying new aspects of tumor biology involving the immunological composition of metastases.

While the work is of merit and results are really sound some concerns avoid its recommendation as it is.

Concerns:

Comment 1: *It would be recommended since the correlation at transcript and protein level was observed regarding to 2093 genes but about 60 proteins, to increase by other means the correlation between RNA and protein (i.e. immunohistochemistry, WB, qPCR) from selected targets.*

Response: Thank you for your thoughtful comments on this manuscript. A strong correlation between RNA/transcript DSP and protein DSP grouped by phenotype was observed throughout these experiments (Figure S3c). IHC has been performed for three prostate specific markers,

AR, PSA and SYP, as a validation of both the transcript and protein data and is included in the supplemental data (Figure S3e). Consistent expression levels between IHC and DSP transcript data were observed (KLK3/PSA, AR and SYP). Some discordance between IHC, particularly the SYP antibody, and protein DSP was observed.

Comment 2: *The manuscript would be benefitted if the authors will be able to discuss potential therapeutic targets analyzed by DSP and found to be dysregulated (or not) and could aid in the therapy of patients with no response to immune-based therapy to provide further information of the usefulness of DSP to provide data to manage patients.*

Response: A key finding of the present study demonstrates that the commonly targeted immune checkpoint proteins – PD-1, PD-L1 and CTLA4 are absent, or expressed at very low levels in nearly all prostate cancer metastases. This finding, coupled with the overall low mutation burden found in PC metastases should give pause to using currently approved ICB therapy to treat these patients. The two therapeutic targets we have nominated, B7-H3 and TIM-3, by virtue of the consistent and high expression, are worthy of more study to determine therapeutic benefit. A more detailed discussion on the detection of immune molecules (TIM-3 and B7-H3) in this dataset and their potential role in treatment strategies has been added to the manuscript (page 13-14). The next stage of analyses will focus on the peri-tumoral regions and tumor microenvironments and hopefully additional insights can be gained with respect to therapy.

Comment 3: *Regarding ARv7 expression in those tumors with discordance between Arv7 status measured by DSP and RNAseq, have the authors considered to verify by other means (i.e. amplification of Arv7 transcript by PCR) whether DSP or RNAseq status fail?. Depending on the results, this will reinforce the usefulness of DSP to detect alternative splicings.*

Response: As suggested, immunohistochemistry (IHC) was performed for AR-V7. IHC was quantitated using H-score as previously described with a specific AR-V7 antibody at a 1:2000 dilution (PMID: 30334814). There was a significantly positive correlation between IHC H-score and DSP $r = 0.54$, $p < 0.0001$ (Fig S3g). We conclude that assessment of AR-V7 splice variant status can be determined by DSP within a larger more comprehensive analysis platform, compared to assays for individual proteins (IHC) or transcript (RT-PCR; ISH; etc).

We note discrepancies did exist between IHC vs DSP vs RNAseq. Part of this may result from the fact that different ‘pieces’ of the tumor were used for each assay, with the bulk RNAseq performed on a tumor piece adjacent to the piece used for the TMA construction. Further, the TMA section used for DSP was ~50 μm ‘distant’ from the TMA section used for IHC (10 sections in-between). In one sample, 17-081, RNAseq was negative for AR-V7 whereas IHC was positive, and in another metastasis from this same patient RNAseq and IHC were both positive for AR-V7. This may represent the heterogeneity of V7 in metastases as has been previously published (PMID: 30334814) and is also present in this analysis (Fig S3f). In three metastases, DSP and RNAseq were positive for AR-V7 but IHC was negative. The reason for this discrepancy is not clear but may be due to antibody sensitivity at low AR-V7 protein levels. The manuscript has been updated on pages 10 and 13 to reflect this discussion.

REVIEWER COMMENTS

Reviewer #1 (Remarks to the Author):

All previous points raised have been adequately addressed by the authors.

Reviewer #2 (Remarks to the Author):

The authors added a statistical measure of homogeneity, however without reference to inter-tumoral pairs it is not possible to assess intra-tumoral homogeneity. In other words, the authors should compare the 82% to the value obtained when considering pairs across patients. Also, the authors should also add the intra-tumor pair values for each tumor individually. Additionally, it is odd to call this a probability when the exact value can be calculated by comparing the pairs. In other words, the authors are discussing frequencies not probabilities.

For Comment 3: we meant to choose one criterion and stick with it for all of the Figure 4d sections (currently, each one is sorted separately).

In Comment 5, the authors claim to have only made a qualitative statement, but the plot and associated text are clearly quantitative: "ROIs from regions spatially distant from the PanCK positive cells progressively lost AR activity and gained a gene expression program associated with transdifferentiation to a NE phenotype such as SOX2 and MYCN as well as increased FGF/FGFR/MAPK activity (Figure 5c,e,f)." Thus if a statistical test is absent then it cannot be concluded that "PanCK positive cells are progressively lost AR activity". The authors are showing graphs in Figure 5 so they cannot make the case that the data is qualitative as opposed to quantitative.

With regards to B7-H3, the authors should make clear in the text what is already known and what is a novel finding from their work. Although they have added the suggested citation, they do not adequately explain their results in the context of the literature.

Reviewer #3 (Remarks to the Author):

In the revised version of the manuscript the authors have completed next generation sequencing data for the DSP protein experiments, updating methods section, figures and supplementary data. Therefore, the manuscript has been strengthened not only by the inclusion of these experiments but also for the correct responses to the reviewers that also included new material correctly depicted in the revised version of the manuscript. In this sense, the authors have completely addressed the concerns raised by this reviewer, and thus, the manuscript is recommended for publication.

RE: # NCOMMS-20-27107
Nature Communications
Editorial Staff and Reviewers

REVIEWER COMMENTS – Point by point response to manuscript submission # NCOMMS-20-27107A

Reviewer #1 (Remarks to the Author):

All previous points raised have been adequately addressed by the authors.

Reviewer #3 (Remarks to the Author):

In this sense, the authors have completely addressed the concerns raised by this reviewer, and thus, the manuscript is recommended for publication.

Reviewer #2 (Remarks to the Author):

Comment 1. *The authors added a statistical measure of homogeneity, however without reference to inter-tumoral pairs it is not possible to assess intra-tumoral homogeneity. In other words, the authors should compare the 82% to the value obtained when considering pairs across patients. Also, the authors should also add the intra-tumor pair values for each tumor individually. Additionally, it is odd to call this a probability when the exact value can be calculated by comparing the pairs. In other words, the authors are discussing frequencies not probabilities.*

Response. We have completed the requested analysis and provided the data in the Results section. We have also removed the word ‘probability’ from the description of the results.

Line 163. Overall, there was high concordance in the phenotype call within a given patient with 82% of randomly sampled pairs of tumor ROIs from the same patient having the same phenotype classification (**Figure 3a**). In contrast, only 54% of ROIs were phenotypically concordant when randomly comparing ROIs across all tumors and all patients.

Line 212: Overall, there was high intra-tumor concordance for the transcript-defined tumor phenotypes (**Figure 4a**) with 96% of randomly sampled ROI pairs from the same tumor having the same phenotype classification.

We are not clear what the reviewer means by this request: “Also, the authors should also add the intra-tumor pair values for each tumor individually” In addition to the data described above, these results are readily observed in Figure 4a in the row “DSP class” which shows the classification of each ROI. Nearly all pairs are concordant as noted above – 96% intra-tumor ROI concordance.

Comment 2. *For Comment 3: we meant to choose one criterion and stick with it for all of the Figure 4d sections (currently, each one is sorted separately).*

Response 2. We have re-constructed the figures in 4d and ordered all plots based on AR score.

The addition to the figure legend is: “The NE and CCP plots retain the ordering by the AR score as shown in the top plot.”

Comment 3. *In Comment 5, the authors claim to have only made a qualitative statement, but the plot and associated text are clearly quantitative: “ROIs from regions spatially distant from the PanCK positive cells progressively lost AR activity and gained a gene expression program associated with transdifferentiation to a NE phenotype such as SOX2 and MYCN as well as increased FGF/FGFR/MAPK activity (Figure 5c,e,f).” Thus if a statistical test is absent then it cannot be concluded that “PanCK positive cells are progressively lost AR activity”. The authors are showing graphs in Figure 5 so they cannot make the case that the data is qualitative as opposed to quantitative.*

Response 3. We have removed the word ‘progressively’ and reframed the description of the results. We have also included new figures that quantitate the expression of several representative genes in the spatially distinct ROIs that serve to support the conclusion that ROIs distant from the CK+ region of the tumor have lower expression of the AR and AR-regulated genes and increased expression of genes associated with cell plasticity and NE differentiation such as EZH2.

Line 245: ROIs from regions spatially distant from the PanCK positive cells classified as AR-/NE+ or AR-/NE-, with lower expression of AR and AR regulated genes such as TMPRSS2, and increased expression of genes associated with transdifferentiation, cell plasticity and proliferation such as EZH2 and Ki67 (Figure 5c-i).

Comment 4. *With regards to B7-H3, the authors should make clear in the text what is already known and what is a novel finding from their work. Although they have added the suggested citation, they do not adequately explain their results in the context of the literature.*

Response 4. The literature on B7-H3 is extensive and a PubMed query lists 534 manuscripts. We are not clear as to what the reviewer has in mind with respect to ‘*what is already known*’. We have cited several studies that describe the function of B7-H3 that is already known. The present study does not have a focus on B7-H3, and our point here is that among checkpoint molecules, B7-H3 is among the most highly expressed in metastatic prostate cancer and particularly at higher levels than other immune checkpoints such as CTLA4, PD-L1, etc. This latter point has not been specifically addressed in any publication we are aware of. We identified another manuscript that reported high B7-H3 in prostate cancer bone metastases. We have cited this study and state that our findings confirm this prior finding and in addition our study demonstrates B7-H3 expression in metastases to other organ sites. To be explicit, we have added wording:

“...the current study suggest the capability of DSP to identify potential therapeutic targets and support further studies of these immune modulatory proteins as therapeutic targets in mPC.”

REVIEWERS' COMMENTS<

Reviewer #2 (Remarks to the Author):

All of my points raised have been adequately addressed by the authors.

RE: # NCOMMS-20-27107
Nature Communications
Editorial Staff and Reviewers

**REVIEWER COMMENTS – Point by point response to manuscript submission #
NCOMMS-20-27107B**

Reviewer #2 (Remarks to the Author):

All previous points raised have been adequately addressed by the authors.